# Visual Structures Help Visual Reasoning: Addressing the Binding Problem in LVLMs

**Amirmohammad Izadi** [*], **Mohammad Ali Banayeeanzade** [*], **Fatemeh Askari,**
**Ali Rahimiakbar, Mohammad Mahdi Vahedi, Hosein Hasani,**
**and Mahdieh Soleymani Baghshah**

Department of Computer Engineering
Sharif University of Technology

## Abstract

Despite progress in Large Vision-Language Models (LVLMs), their capacity for visual reasoning is often limited by the *binding problem*: the failure to reliably associate perceptual features with their correct visual referents. This limitation underlies persistent errors in tasks such as counting, visual search, scene description, and spatial relationship understanding. A key factor is that current LVLMs process visual features largely in parallel, lacking mechanisms for spatially grounded, serial attention. This paper introduces Visual Input Structure for Enhanced Reasoning (VISER), a simple, effective method that augments visual inputs with low-level spatial structures and pairs them with a textual prompt that encourages sequential, spatially-aware parsing. We empirically demonstrate substantial performance improvements across core visual reasoning tasks, using only a single-query inference. Specifically, VISER improves GPT-4o performance on visual search, counting, and spatial relationship tasks by 25.0%, 26.8%, and 9.5%, respectively, and reduces edit distance error in scene description by 0.32 on 2D datasets. Furthermore, we find that the visual modification is essential for these gains; purely textual strategies, including Chain-of-Thought prompting, are insufficient and can even degrade performance. VISER underscores the importance of visual input design over purely linguistically based reasoning strategies and suggests that visual structuring is a powerful and general approach for enhancing compositional and spatial reasoning in LVLMs.

## 1 Introduction

Large Language Models (LLMs) have recently achieved remarkable progress, matching or even surpassing human performance across a range of complex tasks [1–5]. Despite significant advancements, Large Vision-Language Models (LVLMs) still lag behind LLMs in core reasoning capabilities. While LLMs show strong symbolic and abstract reasoning, LVLMs struggle with essential aspects of visual understanding. They frequently miscount objects in cluttered or overlapping scenes [6, 7], perform poorly in spatial reasoning (e.g., identifying left of or above) [8], and often fail to accurately locate or recognize salient targets during visual search [9]. Moreover, compositional and relational reasoning, where understanding critically depends on structured relationships between objects, remains a major challenge [6]. These limitations indicate that LVLMs, despite their power, lack key mechanisms for structured visual reasoning. Consequently, their performance in these areas does not yet match that of language-only models in comparable text-based reasoning tasks.

---

[*]Equal contribution.

39th Conference on Neural Information Processing Systems (NeurIPS 2025).

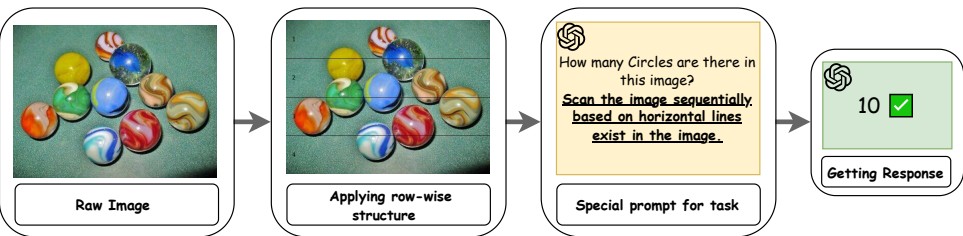

Figure 1: The input image is augmented with low-level visual structure using three horizontal lines, optionally accompanied by row annotations on the left side of the image. A corresponding textual prompt (*Scan the image sequentially based on horizontal lines*) is appended to encourage the model to adopt a spatially guided, sequential parsing strategy.

The limitations in visual reasoning tasks can largely be attributed to a fundamental challenge known as the *binding problem* [9]. Originating from cognitive science and neuroscience, the binding problem refers to the difficulty of correctly associating visual features (like shape and color) and spatial properties (such as location and orientation) with the right objects in a scene. It arises when a system must handle multiple entities simultaneously, leading to interference and misattribution of features [10–12]. In the context of LVLMs, the binding problem manifests as errors in compositional understandingmodels may recognize individual objects correctly but confuse or conflate their attributes. These issues are particularly prevalent in complex, cluttered scenes and persist across a range of model architectures and benchmarks. As a result, reliable multi-object reasoning and generalization of visual concepts to novel settings remain difficult for current LVLMs.

Inspired by neuroscientific studies on human scene perception, we propose **V**isual **I**nput **S**tructure for **E**nhanced **R**easoning (VISER), a novel approach to improve LVLM binding performance. Our approach augments visual inputs with low-level spatial structures, such as horizontal or grid lines, complemented by appending a textual instruction to the input prompt (see Figure 1 as an example). The motivation behind our strategy is to encourage sequential processing in LVLMs guided by these structures in the inputs. Our method can be viewed as a visual analogue of Chain-of-Thought (CoT) prompting[13] for LLMs, as it injects an inductive bias directly into the visual input to implicitly decompose the problem and guide the models reasoning.

Despite its simplicity, VISER yields significant performance improvements. The method is general and input-agnostic, requiring no additional processing for specialized image manipulation. We empirically show that both the visual structure and the textual part of our method are necessary and complementary. In particular, textual prompts for sequential scanning alone prove insufficient to resolve binding failures, and even attempts to elicit more structured reasoning through complex prompting strategies, such as CoT, can paradoxically degrade performance on these visual tasks [14]. In contrast, the inclusion of visual cues stimulates more structured internal processing in LVLMs and promotes spatial parsing mechanisms, enhancing the models ability to bind features accurately. These findings highlight the potential of targeted visual input modifications to improve graphical reasoning and compositional understanding in LVLMs. In summary, the main contributions of this paper are:

- We propose VISER, a novel method that combines explicit visual scaffolding (e.g., horizontal lines) with targeted textual prompts to improve feature-object binding in LVLMs.
- Through extensive experiments, we demonstrate that our method significantly enhances LVLM performance across a range of core visual reasoning tasks, including visual search, counting, scene description, and spatial relationship understanding. For example, on 2D benchmark, VISER improves GPT-4o performance on visual search, counting, and spatial reasoning by 25.0%, 26.8%, and 9.5% (absolute), respectively, and reduces the edit-distance error in scene description by 0.32.
- Our results establish that purely textual interventions are insufficient to overcome binding limitations, and explicit visual manipulation is crucial for this purpose.
- The proposed approach achieves these improvements efficiently, operating within a single query and incurring negligible computational overhead. Its effectiveness, simplicity, and generality (both task- and model-agnostic) distinguish our method from multi-query or agentic approaches that rely on additional processing or tool use.

## 2 Related Work

**LLM and LVLM Reasoning.** While early LLMs were regarded as next-token predictors with little reasoning ability [15, 16], recent work has begun to extensively challenge this problem [17–20]. Todays state-of-the-art models deliver high performance across many tasks, including graduate-level question answering [21] and competitive programming. The improved reasoning abilities of LLMs have naturally sparked interest in reasoning for LVLMs. Initial approaches include Visual Chain-of-Thought [22], knowledge graph integration [23], and tree search [24]. Although these techniques show promise, LVLMs still struggle with tasks such as counting, visual search, scene description, and spatial reasoning, as demonstrated by benchmarks like EMMA [14] and SPACE [25]. Our work aims to incorporate an explicit reasoning pathway into the visual component of the model to improve its performance on these challenging tasks.

**Binding Problem.** The binding problem provides a key explanation for the limitations of state-of-the-art LVLMs in reliably linking visual features to their corresponding objects [26, 27]. Prior work attributes performance issues in tasks such as counting, visual search, and scene description to failures in this featureobject binding in LVLMs [9]. Recent neuroscience research has shown that grid-based frameworks enhance visual recognition memory [28] and improve face recognition performance [29]. Additionally, humans reduce interference by detecting individual objects iteratively [30, 10], and grid structures facilitate movement-based object recognition [31], thus providing insight into how the human brain can deal with the binding problem. Inspired by these insights, we use horizontal lines in images to enhance model reasoning, both with synthetic images and real-world datasets, such as the Learning To Count Everything benchmark [32].

**Agentic LVLMs.** An emerging line of research treats LVLMs as autonomous tool users that can invoke external modules or produce executable artefacts to compensate for perceptual or reasoning gaps. LVLM-COUNT [33] decomposes enumerative queries into sub-counts solved by a specialized counting utility, yielding gains on counting benchmarks. Visual Sketchpad [34] endows multimodal LMs with a drawable canvas, letting them sketch auxiliary lines and marks. Code execution agents such as ViperGPT [35] translate natural-language questions into Python scripts that orchestrates off-the-shelf vision tools, achieving state-of-the-art results on compositional visual queries. Toolformer [36] shows that language models can self-supervise decisions about when and how to call external APIs. Although such tool-using approaches improve task performance, they do not necessarily enrich models' intrinsic reasoning capabilities. In contrast, our work embeds an explicit reasoning pathway directly into the visual input to overcome these limitations.

## 3 Proposed Method

The binding problem, a fundamental challenge in cognitive science and neuroscience, concerns how perceptual systems correctly associate features like color, shape, and location with the right objects in complex visual scenes [10, 12]. When multiple objects share representational resources, the system can produce *illusory conjunctions*, i.e., erroneous confusion of features from different objects (e.g., mistakenly perceiving a red square when shown a red circle and a green square) [11]. These misbindings reveal the difficulty of forming coherent, object-specific representations from distributed visual inputs; their analysis provides a useful framework for evaluating visual reasoning systems.

Neuroscientific studies indicate that the human visual system operates in two distinct modes: a fast, imprecise parallel processing (often likened to System 1 processing) and a more accurate sequential attention (System 2 reasoning) [37]. While parallel processing enables rapid scene understanding, it is prone to errors, particularly in cluttered environments where the demands for feature binding are high. Given sufficient time, humans transition to sequential attention, which allows for more meticulous feature binding and reduces errors. In contrast, current LVLMs predominantly process visual inputs largely in parallel. This approach, while beneficial for compositional representation learning and generalization, inherently risks interference and binding errors if feature-to-object linkages are not precisely managed [27].

To address the binding problem in LVLMs, we propose a lightweight, model-agnostic approach that enhances structured visual reasoning through minimal visual scaffolding and spatially grounded prompting. Inspired by cognitive science findings on serial attention and spatial structuring, our method guides LVLMs to process visual information region-by-region and reduces interference ef-

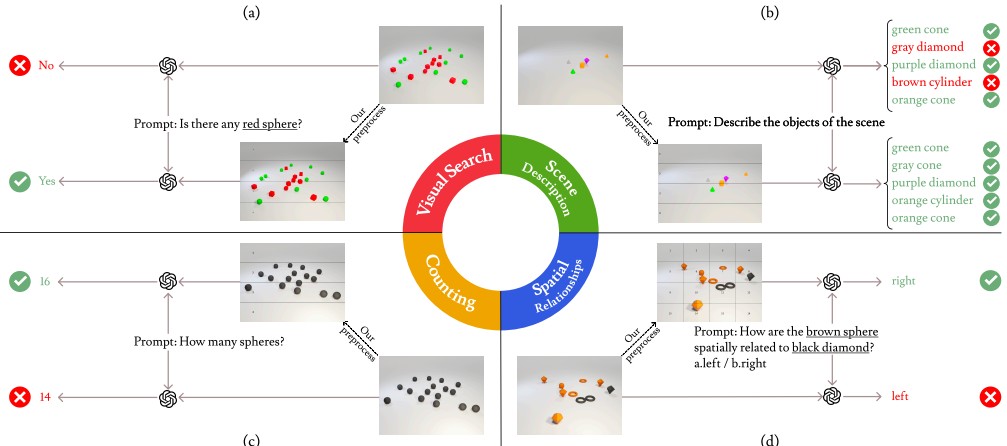

Figure 2: A brief summary of tasks with one example of synthetic data along the specific prompt for each task. (a) Visual Search, (b) Scene Description, (c) Counting, and (d) Spatial Relationship.

fects from parallel processing, a known contributor to binding errors in human and machine vision. This approach introduces negligible computational overhead, requires no task-specific fine-tuning or multi-query visual processing, and comprises two main components, which are illustrated with an example in Figure 1 and explained below:

**Visual Structuring with Horizontal Lines**   We augment each input image with $n$ equidistant horizontal lines, creating $n + 1$ horizontal segments. Each segment is annotated with a numerical label (1 through $n + 1$) placed sequentially from top to bottom. These lines serve as visual anchors, promoting localized attention within each region to reduce cross-object interference and improve feature-object binding. Unlike dense grid-based methods that may obscure content, our minimalist design preserves clarity while providing sufficient spatial guidance. The value of $n$ is set to 3 in our experiments.

**Sequential Scanning Prompt**   To align the models attention with the visual scaffold, we prepend a fixed instruction, specifically: *"Scan the image sequentially based on horizontal lines exists in the image."* This prompt guides the model to adopt a structured, row-wise processing strategy, encouraging a systematic evaluation of the image content. For task-specific adaptation (e.g., counting or spatial reasoning), we augment the base prompt with additional instructions (see Appendix C).

The use of manually added artifacts and the emphasis on sequential, region-based processing may also align with the discussion of feature entropy in [9]. By partitioning the image into distinct segments, our method encourages focused processing on smaller subsets of objects. This can simplify the distinction between objects within each segment by reducing the concurrent feature load, an effect pertinent to managing feature diversity (or entropy) as detailed therein. Such simplification is a key factor contributing to improved visual reasoning, particularly in counting.

## 4   Experiments

### 4.1   Setup

**Datasets**   To evaluate VISER, we use both synthetic and natural datasets. A Binding Problem Generator [9] produces synthetic data that can be controlled in two and three dimensions and can incorporate a variable number of objects. Additionally, we benchmark on two real-world tasks: Learning To Count Everything [32] and the Spatial Reasoning [38] datasets.

**Models**   We evaluated our proposed method across a carefully chosen mix of closed-source and open-source LVLMs. Specifically, we include two leading closed-source systemsOpenAIs GPT-4o [39] and Anthropics Claude3.5-sonnet [40]selected for their cutting-edge capabilities and solid performance on multimodal benchmarks. For GPT-4o, we evaluate both our method and Chain-of-Thought (CoT) prompting [13] to enable a direct comparison within the same architecture. Additionally, we benchmark two state-of-the-art open-source models: Qwen2.5-VL-7B-Instruct [41] and LLaMa4-scout-17b-16e-Instruct [42]. Beyond these baselines, we assess the benefits of targeted fine-tuning by incorporating Mulberry [24], a Qwen2.5VL-7B variant refined for enhanced multi-

Table 1: Performance comparison on the counting task using models GPT-4o, Claude3.5-sonnet, LLaMa4, and Qwen2.5-VL, evaluated with accuracy (%) . The comparison is conducted across 2D and 3D datasets with varying numbers of objects in the scene, as well as a natural image dataset. Results are shown for base models and VISER.

| Dataset | Objects | GPT-4o | | Claude-sonnet | | LLaMa4 | | Qwen2.5-VL | |
|---|---|---|---|---|---|---|---|---|---|
| | | Baseline | VISER | Baseline | VISER | Baseline | VISER | Baseline | VISER |
| 2D | 10 | 42.00 | **65.00** | **26.00** | **26.00** | 23.00 | **31.00** | 18.00 | **41.00** |
| | 12 | 12.00 | **40.00** | 5.00 | **15.00** | 3.00 | **20.62** | 1.00 | **45.00** |
| | 14 | 1.00 | **34.00** | 5.00 | **9.00** | 0.00 | **23.47** | 1.00 | **13.00** |
| | 16 | 3.00 | **39.00** | 3.00 | **6.00** | 11.00 | **23.00** | 1.00 | **59.00** |
| | 18 | 1.00 | **29.00** | **6.00** | 4.00 | 0.00 | **9.28** | 0.00 | **10.00** |
| | 20 | 13.00 | **26.00** | **6.00** | 4.00 | 1.00 | **7.22** | 14.00 | **77.00** |
| | **Avg** | 12.00 | **38.83** | 8.50 | **10.67** | 6.33 | **19.10** | 5.83 | **40.83** |
| 3D | 10 | 52.00 | **62.00** | **56.00** | 54.00 | 32.00 | **60.42** | 20.83 | **66.00** |
| | 12 | 12.00 | **38.00** | 26.00 | **28.00** | 0.00 | **34.69** | 8.16 | **20.00** |
| | 14 | 6.00 | **14.00** | 14.00 | **24.00** | 12.00 | **38.00** | 4.08 | **24.00** |
| | 16 | 16.00 | **24.00** | 8.00 | **14.00** | 40.00 | **44.00** | 6.00 | **16.00** |
| | 18 | 2.00 | **18.00** | 0.00 | **8.00** | 0.00 | **18.75** | 0.00 | **14.00** |
| | 20 | 2.00 | **30.00** | 0.00 | **4.00** | 0.00 | **22.45** | 12.00 | **20.00** |
| | **Avg** | 15.00 | **31.00** | 17.33 | **22.00** | 14.00 | **36.39** | 8.51 | **26.67** |
| Natural | | 29.82 | **35.65** | 2.09 | **6.42** | 29.22 | **31.65** | **18.91** | 17.29 |

modal reasoning. This breadth of model selection allows us to rigorously assess the generality and robustness of our approach across diverse architectures, training scales, and access constraints.

**Evaluation Metrics**   We use three metrics adopted in visual reasoning research: accuracy, harmonic mean, and edit distance. Accuracy, used for counting and spatial relationship tasks [14], measures the proportion of correct predictions. Harmonic mean evaluates visual search by balancing performance across visible and invisible object detection [9], ensuring that high scores require consistent performance on both subtasks. Edit distance (Levenshtein distance) computes the minimum number of insertions, deletions, or substitutions needed to transform a model-generated scene description into the reference [9], offering a fine-grained measure of binding precision in generated text.

In addition to the main experiments, ablation studies are provided in Appendix A, where we isolate the effect of each component of VISER and evaluate various hyperparameters across different models. Specifically, we assess variants that use only the visual scaffolding or only the sequential prompt (see Appendix A.1), as well as different numbers of structural lines. Moreover, we investigate the impact of other design choices, such as the number and thickness of lines (Appendix A.2). These analyses confirm that both components contribute to the overall performance gains and that VISER remains effective across a reasonable range of hyperparameter settings. Finally, in Appendix B.3, we show the significance of the proposed method by applying a binomial sign test on the results.

## 4.2   Counting

Counting involves determining the number of specific objects within a scene. Although it may appear simple, accurate enumerationespecially when multiple object types or attribute variations are involvedrelies heavily on effective feature binding. Each object must be individuated, its defining features correctly associated, and then counted as a distinct instance. Failures in binding can lead to omissions, double-counting, or misclassification. Similar to human rapid estimation under time constraints, LVLMs show capacity limits in counting. Performance often worsens as object numbers grow, or when high object similarity (low feature heterogeneity) increases the interference and binding errors.

An instance of this task is demonstrated in Figure 2-(c); To evaluate our method, we generated 2D and 3D images, using the synthetic data set described above, each containing between 10 and 20 instances of the target object to increase complexity. We measure performance using counting accuracy; the results are summarized in Table 1. As presented, our countingspecific prompting yields substantial gains on synthetic benchmarks: in 2D scenes, GPT-4o jumps from 12.00% to 38.83%,

Table 2: Harmonic mean comparison for the Visual Search task, evaluating VISER against base models across GPT-4o, Claude3.5-sonnet, LLaMa4, and Qwen2.5-VL on 2D and 3D datasets with varying numbers of objects.

| Dataset | Objects | GPT-4o | | Claude-sonnet | | LLaMa4 | | Qwen2.5-VL | |
|---------|---------|----------|-------|----------|-------|----------|-------|----------|-------|
| | | Baseline | VISER | Baseline | VISER | Baseline | VISER | Baseline | VISER |
| 2D | 20 | 0.71 | **0.91** | 0.55 | **0.82** | 0.00 | **0.06** | 0.31 | **0.43** |
| | 30 | 0.50 | **0.80** | 0.54 | **0.70** | 0.00 | 0.00 | 0.28 | **0.37** |
| | 40 | 0.25 | **0.64** | 0.18 | **0.59** | 0.00 | 0.00 | 0.14 | **0.23** |
| | 50 | 0.46 | **0.55** | 0.10 | **0.55** | 0.00 | 0.00 | 0.47 | **0.56** |
| | **Avg** | 0.48 | **0.73** | 0.34 | **0.66** | 0.00 | **0.02** | 0.30 | **0.40** |
| 3D | 20 | **1.00** | **1.00** | **1.00** | **1.00** | **0.82** | 0.80 | 0.33 | **0.44** |
| | 30 | **0.96** | **0.96** | 0.88 | **0.91** | 0.43 | **0.60** | 0.15 | **0.18** |
| | 40 | 0.89 | **0.94** | 0.72 | **0.80** | **0.59** | 0.58 | 0.00 | **0.15** |
| | 50 | 0.81 | **0.84** | 0.59 | **0.75** | **0.39** | 0.38 | 0.00 | **0.04** |
| | **Avg** | 0.91 | **0.93** | 0.80 | **0.86** | 0.56 | **0.59** | 0.12 | **0.20** |

Claude-sonnet from 8.50% to 10.67%, LLaMa4 from 6.33% to 19.10%, and Qwen2.5-VL surges from 5.83% to 40.83%; in 3D scenes, GPT-4o climbs from 15.00% to 31.00%, Claude-sonnet from 17.33% to 22.00%, LLaMa4 from 14.00% to 36.39%, and Qwen2.5-VL from 8.51% to 26.67%. On natural images, improvements are more modestGPT-4o from 29.82% to 35.65%, Claude-sonnet from 2.09% to 6.42%, LLaMa4 from 29.22% to 31.65%and Qwen2.5-VL sees a slight drop (18.91% to 17.29%). Together, these results underscore the effectiveness of taskaware prompting in controlled settings while highlighting the remaining challenge of generalizing to realworld imagery.

## 4.3 Visual Search

Visual search task challenges models to locate a target object among distractors, with task difficulty adjusted by the similarity between target and distractor features. In conjunctive search, where the target is defined by a unique combination of features (e.g., a green L-shape) and distractors partially share these features (e.g., red L-shapes, green T-shapes), successful identification depends on accurate feature binding. Models that struggle with binding may exhibit degraded performance as the number of distractors increases or when features are highly confusable, mirroring human performance limitations when serial attentional scanning is prevented [10].

An instance of the visual search problem is presented in Figure 2-(a); we evaluate the proposed method using the synthetic dataset described earlier to generate 2D and 3D scenes containing between 20 and 50 objects of varying complexities. We assessed performance by measuring accuracy in correctly identifying the targets presence or absence, distinguishing between visible searches (the object is explicitly in the scene) and invisible searches (the object is absent). This distinction is important because simpler LVLMs often default to predicting false, inflating their invisibleobject accuracy while failing to detect visible targets. After computing the accuracy for visible and invisible subtasks, we combined them via the harmonic mean to produce a final score.

$$\text{Harmonic Mean} = \frac{2 \cdot \text{Visible} \cdot \text{Invisible}}{\text{Visible} + \text{Invisible}} \quad (1)$$

Table 2 compares our approach against baselines, showing VISER delivers consistent and substantial gains over the baseline across 2D and 3D scenes and for all evaluated LVLMs. In 2D settings, the average harmonic mean for GPT-4o climbs from 0.48 to 0.73, for Claude3.5-sonnet from 0.34 to 0.66, for LLaMa4 from 0.00 to 0.02, and for Qwen2.5-VL from 0.30 to 0.40, with the largest relative improvements occurring as scene complexity increases (e.g., at 40 objects GPT-4o rises from 0.25 to 0.64, and Claude3.5-sonnet rises from 0.18 to 0.59). In 3D scenes, VISER still yields gains to 0.93, and weaker models benefit notably (Claude3.5-sonnet: 0.80  0.86; LLaMa4: 0.56  0.59; Qwen2.5-VL: 0.12  0.20). These results confirm that our strategy not only corrects the always-false bias of simpler LVLMs in invisible-object tasks but also markedly enhances their ability to detect visible targets, achieving robust performance regardless of object count, visibility, or scene dimensionality.

Table 3: Comparison of scene description performance (lower is better; measured as average distance) for VISER versus baseline across GPT-4o and Claude3.5-sonnet on 2D and 3D datasets with varying numbers of objects.

| Dataset | Objects | GPT-4o | | Claude-sonnet | | LLaMa4 | | Qwen2.5-VL | |
|---|---|---|---|---|---|---|---|---|---|
| | | Baseline | VISER | Baseline | VISER | Baseline | VISER | Baseline | VISER |
| 2D | 10 | 0.70 | **0.67** | 4.68 | **3.12** | **2.03** | 2.29 | 4.38 | **4.24** |
| | 15 | 1.79 | **1.48** | 2.89 | **2.24** | **3.19** | 3.99 | 7.71 | **7.35** |
| | 20 | 3.32 | **2.73** | 1.45 | **1.24** | **5.74** | 6.89 | 12.29 | **10.59** |
| | **Avg** | 1.94 | **1.62** | 3.01 | **2.20** | **3.65** | 4.39 | 8.12 | **7.39** |
| 3D | 10 | **4.06** | 4.25 | 5.40 | **5.18** | **5.04** | 5.55 | 7.88 | **6.61** |
| | 15 | **7.29** | 7.48 | 8.56 | **7.95** | **9.39** | 10.60 | 11.56 | **10.69** |
| | 20 | 12.38 | **12.21** | 14.37 | **12.03** | **14.90** | 16.45 | 15.84 | **15.24** |
| | **Avg** | **7.91** | 7.98 | 9.45 | **8.39** | **9.78** | 10.84 | 11.76 | **10.84** |

Table 4: Accuracy (%) comparison of spatial relationship predictions among GPT-4o, Claude 3.5-Sonnet, LLaMA 4, and Qwen2.5-VL on 2D, 3D, and natural image datasets, using both baseline methods and VISER.

| | GPT-4o | | Claude-sonnet | | LLaMa4 | | Qwen2.5-VL | |
|---|---|---|---|---|---|---|---|---|
| | Baseline | VISER | Baseline | VISER | Baseline | VISER | Baseline | VISER |
| 2D | 43.00 | **52.50** | 34.18 | **36.26** | **29.50** | 27.50 | 48.50 | **50.00** |
| 3D | 64.00 | **68.50** | 72.50 | **76.50** | 55.00 | **58.50** | 78.00 | **82.50** |
| Natural | 69.39 | **77.43** | 37.43 | **46.15** | 58.67 | **66.84** | 80.10 | 77.04 |

## 4.4 Scene Description

The scene description task requires models to generate accurate textual narratives that describe the objects, their attributes, and their relationships within an image. This task challenges a LVLM's ability to solve the binding problem. The model must not only detect features (e.g., red, cube, blue, sphere) but also correctly associate them with the corresponding objects in its textual output (e.g., a red cube and a blue sphere, not a blue cube and a red sphere). Errors, such as attribute swapping or incorrectly pairing features with objects, often arise in complex scenes. This typically occurs when multiple objects share some features but differ in others, which inherently challenges precise feature-object binding.

Figure 2-(b) shows an instance of the scene description task; we evaluate a synthetically generated dataset of 2D and 3D scenes, each containing between 10 and 20 objects to vary scene complexity. Model performance is quantified by the edit distance between the generated descriptions and the ground-truth annotations; we show more detail about this in Appendix B. This yields three outcome categories: (1) an exact match, (2) a single-attribute mismatch (either shape or color), and (3) a double-attribute mismatch (both shape and color). The full results are presented in Table 3. The greatest average gains (-0.81 in 2D and -1.06 in 3D) are driven precisely where the task's difficulty peaks, showing that the technique not only endures extra clutter but thrives on it. This confirms that VISER is the most reliable option for reasoning with a high object count.

## 4.5 Spatial Relationship

In this task, the model is expected to identify and verify the relative positions of objects in a scene (e.g., Is the red cube to the left of the blue sphere?). This involves not only correctly binding intrinsic features (e.g., color, shape) to objects but also accurately associating each object with its spatial location, followed by a relational comparison. Failure to bind features properly can lead to misidentification of the objects being queried. Furthermore, if the LVLM cannot accurately associate each identified object with its precise location, its capacity to reason about their relative positions will be compromised. LVLMs often struggle with spatial reasoning due to their inability to maintain distinct, spatially grounded representations of multiple objects simultaneously [8].

An example of the spatial relationship task is presented in Figure 2-(d); To evaluate VISER, we generated 2D and 3D scenes using the synthetic dataset described earlier and posed four-option multiple-choice questions for each scene, asking models to select the answer that best describes the spatial relation between the target objects. The evaluation results are shown in Table 4. As you can

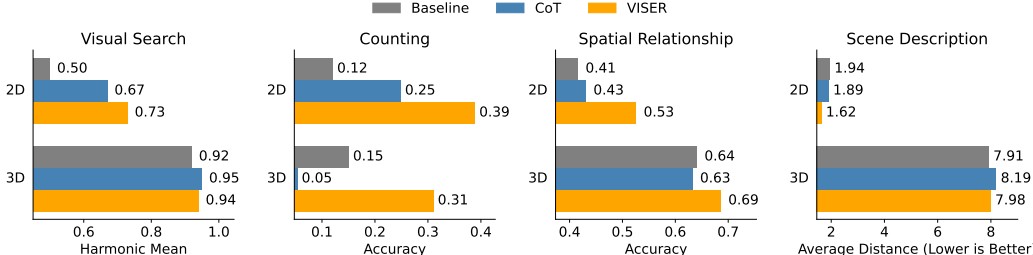

Figure 3: Comparison of VISER with Chain-of-Thought (CoT) prompting on the GPT-4o model across four tasks. Each subplot shows the performance of three methods (Baseline, CoT, and VISER) evaluated using a task-specific metric indicated on the x-axis. Bars are grouped into 2D and 3D datasets.

see, the new results show that VISER yields consistent gains for most models: GPT-4o rises from 43.00% to 52.50% (+9.50%) on 2D, 64.00% to 68.50% (+4.50%) on 3D, and 69.39% to 77.43% (+8.04%) on natural scenes; Claude-sonnet jumps from 34.18% to 36.26% (+2.08%) on 2D, 72.50% to 76.50% (+4.00%) on 3D, and 37.43% to 46.15% (+8.72%) on natural images; LLaMa4 dips 2.00% on 2D but improves 3.50% on 3D and 8.17% on natural; and Qwen2.5-VL gains 1.50% on 2D and 4.5% on 3D yet sees a 3.06% drop on natural scenes.

## 4.6 Chain of Thought

To examine whether purely textual reasoning cues can bridge the binding gap, we pit our visually structured prompt against a CoT baseline. The baseline employs GPT-4o with the canonical cue Lets think step by step [13] appended to each instruction, leaving the image untouched, mirroring standard practice for eliciting step-wise reasoning in language models. Across all four tasks, VISER consistently surpasses both the Baseline model (no reasoning prompt) and the CoT variant in 2D and 3D scenarios (Figure 3). Crucially, the gap is far more pronounced on the 2D datasets. Flattened scenes crowd objects into a single plane, intensifying cross-object interference and making accurate binding harder. In these cluttered 2D settings, the lightweight spatial scaffold provides the model with an external, row-by-row traversal plan, producing markedly larger gains than CoT; in 3D scenes, the relative advantage narrows but persists. These results underline that verbal reasoning alone is not enough. A CoT prompt can encourage the model to explain its answer, but it cannot repair a noisy, globally pooled image embedding. Once early tokens are conditioned on entangled visual features, every subsequent step inherits the same misbindings. The scaffold, by contrast, pre-structures the visual input so that each attended region contains fewer competing objects, allowing the language model to reason over cleaner, localized representations. Thus, reliable multimodal reasoning emerges only when textual chains of thought are paired with a minimal but crucial spatial guide.

## 4.7 Comparison with Fine-Tuned Models for Visual Reasoning

To evaluate whether our lightweight method can match the benefits of model-level adaptation, we compare it to Mulberry[24]a Qwen2.5-VL variant fine-tuned for visual reasoningand OpenVLThinker[43], which enhances reasoning in a Qwen backbone via reinforcement learning using a GRPO implementation. As illustrated in Figure 4, we evaluate the base Qwen2.5-VL model, VISER, the fine-tuned Mulberry, and OpenVLThinker. VISER consistently improves upon the base model and often matches or outperforms fine-tuned models, despite requiring neither fine-tuning nor access to model internals.

For example, on the 2D-Counting task, VISER achieves a significant boost in performance41% accuracy, compared to 15% by Mulberry and the same result from OpenVLThinker. The only tasks where OpenVLThinker shows an advantage are 2D-Spatial reasoning and 3D-Counting, likely due to its reinforcement learning-based optimization for multimodal reasoning. This suggests that while training-intensive methods can excel in specific domains, they incur substantial computational costs.

In contrast, our training-free paradigm demonstrates broader effectiveness across diverse multimodal reasoning tasks, offering superior scalability and efficiency. These results show that addressing

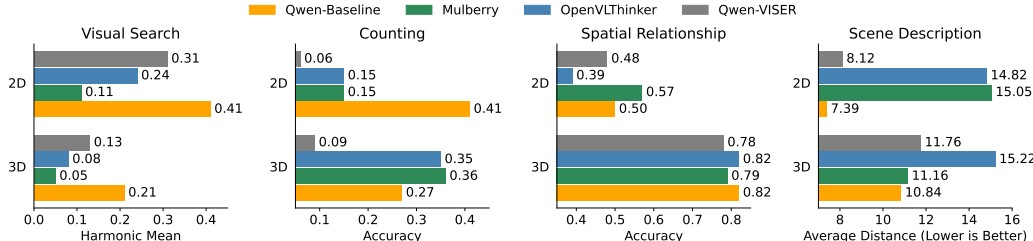

Figure 4: Comparison of VISER with visual reasoningfinetuned model on the Qwen2.5-VL base model across four tasks. Each subplot shows the performance of four methods: Qwen-Baseline (Qwen2.5-VL), Qwen-VISER (VISER applied to Qwen2.5-VL), Mulberry (Qwen2.5-VL finetuned for visual reasoning), and OpenVLThinker (RL-finetuned Qwen2.5-VL), evaluated using task-specific metrics, and results are grouped into 2D and 3D datasets.

Table 5: Performance comparison (Accuracy %) of VISER vs. baseline on the **MMBench** benchmark. Results are reported across three key reasoning categories: Attribute, Logical, and Relation.

| Model | Method | Attribute Reasoning | Logical Reasoning | Relation Reasoning |
|---|---|---|---|---|
| GPT-4o | Baseline | **88.67** | 78.25 | 75.17 |
| | VISER | 88.00 | **82.75** | **81.12** |
| Qwen2.5-VL | Baseline | 80.67 | **80.00** | 82.99 |
| | VISER | **82.33** | **80.00** | **83.33** |

Table 6: Comparison of VISER and the baseline on the **PhysBench** benchmark, covering physical reasoning tasks across four categories: Dynamics, Scene Understanding, Object Relationships, and Object Property. Overall accuracy is also reported. All values are reported as Accuracy (%).

| Model | Method | Dynamics | Scene Understanding | Object Relationships | Object Property | Overall |
|---|---|---|---|---|---|---|
| GPT-4o | Baseline | 35.00 | 33.33 | **75.00** | 55.00 | 55.68 |
| | VISER | **40.00** | **41.67** | **75.00** | **70.00** | **61.37** |
| Qwen2.5-VL | Baseline | **40.00** | **50.00** | 61.11 | 60.00 | 54.55 |
| | VISER | **40.00** | **50.00** | **72.22** | **70.00** | **61.36** |

the binding problem through simple visual scaffolding and spatial prompting (without additional supervision or computational expense) can rival and even surpass state-of-the-art training-based techniques for multimodal reasoning.

## 4.8 Broader Evaluation Across Reasoning Benchmarks

To further validate the robustness and generality of our method, we evaluate it on additional reasoning-focused benchmarks, even in domains where visual structural cues might seem less relevant. These benchmarks include MMBench [44], PhysBench [45], RAVEN [46], and a Visual Analogy task [9]. These evaluations span a broad range of visual reasoning challenges, covering attribute, logical, relational, physical, and analogical reasoning. Results consistently show that our method improves performance across diverse tasks and modalities, demonstrating strong generalization in both abstract and grounded visual reasoning scenarios.

**MMBench** MMbench is a reasoning-focused benchmark composed of tasks in Attribute, Logical, and Relation Reasoning. We include all reasoning tasks in our evaluation. As shown in Table 5, our method surpasses the baseline across most categories using both GPT-4o and Qwen2.5-VL models.

**PhysBench** We also evaluate on PhysBench, a benchmark focused on physical commonsense reasoning with categories including Dynamics, Scene Understanding, Object Relationships, and Object Properties. Table 6 presents results on the validation split. Our approach consistently improves performance across most categories, showcasing its utility in physical reasoning tasks.

**RAVEN**   We evaluate our method on the RAVEN benchmark, which already includes grid lines. Instead of altering the images, we added the prompt: *Scan the image sequentially based on the grid lines present in the image* to encourage the model to leverage existing structure. Using GPT-4o, our method correctly answered 26 out of 140 questions, compared to 19 for the baselinehighlighting that prompting alone, without image alteration, can yield tangible gains when leveraging structural cues.

**Visual Analogy Task**   We evaluate the Visual Analogy task under a unified single-image setup. Using GPT-4o, both the baseline and our method achieve perfect accuracy (100%). For Qwen2.5-VL, our method improves performance from 72.0% to 77.0%, showing benefits in abstract analogical reasoning.

## 5   Discussion

This research investigated the binding problem in LVLMs, a key factor limiting their performance on structured visual reasoning tasks. We introduced VISER, a method that combines simple visual scaffolding, using horizontal lines, with a targeted textual prompt to encourage sequential and spatially aware processing. Our experiments demonstrate that this external intervention leads to substantial and consistent performance improvements across visual search, counting, scene description, and spatial relationship understanding. Notably, VISER operates in a fully input-agnostic manner, applying the same structure across visual inputs from diverse tasks without additional image processing. These results support the hypothesis that explicit low-level general visual structure improves feature-object binding.

A central finding is the critical role of visual structures. While textual prompts alone proved insufficient and sometimes harmful, the addition of visual lines significantly improved reasoning accuracy. These findings suggest that current LVLMs, despite their advanced linguistic capabilities, can benefit greatly from external visual cues that facilitate a more systematic parsing of complex scenes. The visual structure appears to help models approximate a serial attentional process, mitigating the interference and illusory conjunctions when processing multiple objects and their features in parallel. This localized, sequential processing may also reduce representational interference by effectively managing feature entropy within sub-regions of the image, a factor highlighted in [9] as critical for robust visual reasoning. This supports the idea that addressing the binding problem may require interventions at the level of visual input processing, rather than relying solely on linguistic instruction.

VISER offers a practical and efficient intervention, readily applicable due to its simplicity, negligible computational overhead, and single-query operation, without reliance on external tools, model fine-tuning, or architectural changes (see Appendix G).. This highlights visual input structuring as a key strategy for enhancing LVLM reasoning, beyond linguistic instruction. While effective, the current static nature of the visual scaffolding represents a potential area for refinement (see Appendix E for failure cases). For instance, a fixed scaffolding structure might show reduced gains when the interference between lines and objects is too high, or when object clustering limits the spatial separation benefits of the scaffolds. Future research could investigate adaptive visual scaffolding, where cues are dynamically determined by image content or query specifics. Relatedly, ensemble-based approaches that combine diverse scaffolds (e.g., varying line positions, density, or color) could enhance robustness and reduce sensitivity to edge cases. Moreover, developing LVLM architectures that inherently support serial, spatially grounded attention, perhaps inspired by these findings, could offer more integrated and robust solutions to the binding problem. Extending VISER to other visual tasks, such as hallucination detection and mitigation, also represents a promising direction.

Last but not least, we emphasize that improved visual reasoning in AI systems can benefit a wide range of applications, from assistive technologies and robotics to education and healthcare. However, as LVLMs become more capable, their potential misuse (for instance, in surveillance, automated decision-making, or misinformation) also grows. Our approach does not require model retraining, making it widely accessible, but this also implies that such modifications could be adversarially deployed in black-box systems without transparency or oversight. To mitigate risks, we advocate for responsible use of these techniques in alignment with accountability and transparency guidelines. Future research should profoundly study the unintended biases that scaffolding might introduce in tasks other than spatial reasoning, especially when used in safety-critical applications.

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

# A  Ablation study

## A.1  Visual Structures

In our ablation study, we assess multiple visual-structuring strategies and their impact on task performance using the GPT-4o [39] model. All experiments are conducted on a consistent subset of our 2D synthetic dataset [9], with varying object counts per scene. The results in Table 7 show the average performance across these scenes.

Table 7: Comparison of VISER and other ablation methods on GPT-4o in 2D scenes across multiple tasks.

| Method | Scene Desc. (Edit Dist.) | Counting (Accuracy) | Visual Search (Accuracy) | Spatial Rel. (Accuracy) |
|---|---|---|---|---|
| Baseline | 1.81 | 16.85 | 0.48 | 39.00 |
| Row | **1.61** | **31.55** | 0.73 | 44.50 |
| Column | 2.00 | 16.68 | **0.76** | 43.50 |
| Grid | 2.43 | 17.40 | 0.72 | **51.50** |
| RowNoRowNum | 1.71 | 19.14 | 0.70 | 41.00 |

The baseline configuration prompts the model to describe the image without any additional structuring cues. As shown, this baseline yields the lowest or near the lowest accuracy in counting and visual search, indicating that unstructured prompts are insufficient for supporting systematic reasoning.

Among the structured variants, the Grid-based layout consistently performs the worst, particularly in the scene description task, likely due to excessive spatial fragmentation hindering its interpretation of coherent groupings. The Column-based layout also underperforms in both scene description and counting tasks compared to horizontal-based layouts, suggesting that vertical splitting is suboptimal for global scene comprehension. Interestingly, it achieves a small gain in visual search, where object-level localization is more critical than spatial integration.

In contrast, our row-based horizontal structuring, which explicitly divides the image into horizontal bands, leads to improved performance across most tasks. This design encourages the model to scan the scene in a consistent, line-by-line fashion. Furthermore, removing numerical indices from rows (VISERNoRowNum) results in a substantial drop in accuracy, highlighting the value of explicit ordering cues in guiding sequential attention.

For the spatial relation task, we chose a grid because the answer space is symmetric across four directions (left, right, above, below), making the grid a more natural fit. However, as shown in Table 7, horizontal lines also perform well in this task. Specifically, for the synthetic spatial relation task with GPT-4o, accuracy improves from 39.00% (baseline) to 44.50% (row method) and 51.50% (grid method). Similarly, in the natural spatial relation task, accuracy increases from 69.39% to 76.92% and 77.43%, respectively. This demonstrates that our simple row-based method is also effective for spatial relations, but the grid method yields the best results.

These results collectively underscore the importance of structured, order-aware visual annotations in enabling LVLMs to reason more effectively over complex spatial arrangements.

## A.2  Hyperparameters

In this section, we examined the effect of key scaffold parameters, focusing on the number and thickness of horizontal lines used to segment the scene. The initial configuration applied three horizontal lines of 1 px thickness for all tasks, following the heuristic that too few lines (e.g., 12) yield overly coarse partitions with limited benefit, whereas too many lines introduce visual clutter and potential interference artifacts. To systematically assess this, we varied the number of lines (112; Figure 5) and the line thickness (15 px; Figure 6) across three tasksCounting, Visual Search, and Scene Descriptionusing two representative models: GPT-4o (closed-source) and Qwen2.5-VL (open-source). Each configuration was evaluated on 300 samples per task.

Across both models, performance dropped at the extremes but remained stable in the 26 line range. A similar pattern was observed for line thickness, with consistently strong results up to 5 px. Task-specific effects also emerged: in Scene Description, increasing the number of lines often raised the edit distance (lower is better), indicating that excessive segmentation can hinder coherent free-form output. Overall, these results validate our original heuristic and show that the method is robust to

moderate changes in scaffold parameters. While the optimal configuration can vary slightly by task or model, substantial gains are achievable without extensive hyperparameter tuning.

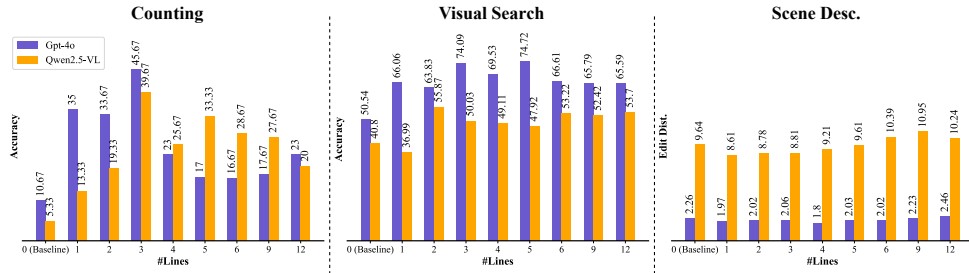

Figure 5: Performance of GPT-4o and Qwen2.5-VL across different tasks (Counting, Visual Search, and Scene Description) with varying numbers of horizontal lines in the input. Accuracy is reported for Counting and Visual Search, while edit distance is used for Scene Description. Baseline represents performance with no horizontal lines.

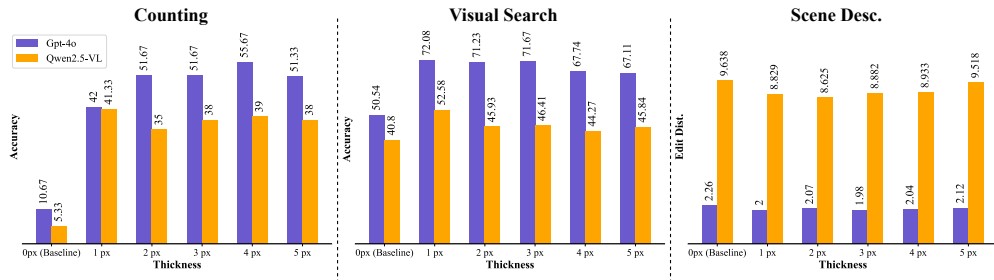

Figure 6: Performance of GPT-4o and Qwen2.5-VL across different tasks with varying line thicknesses (in pixels) introduced by VISER. Each column represents results for a specific line width, ranging from 1 px to 5 px. Accuracy is reported for the Counting and Visual Search tasks, while lower edit distance indicates better performance for the Scene Description task. The Baseline corresponds to performance without added lines (i.e., line thickness of zero).

## A.3 Imaginary Lines

To evaluate the impact of prompting the model to imagine auxiliary structures (e.g., horizontal lines) without rendering them, we introduced an imaginary-baseline variant. This baseline was prompted to scan the image row by row using imaginary linessimilar to the setup illustrated in Figure 27but without explicitly drawing them. As shown in Table 8, VISER outperforms the imaginary-baseline across all tasks: achieving an 8.5% gain in spatial relation accuracy, a 13.31% improvement in counting, a 0.12 reduction in edit distance for scene description, and a 5.00% increase in harmonic mean accuracy for visual search.

Table 8: Performance of the baseline when prompted to imagine horizontal lines (as in Figure 27), evaluated across tasks using the GPT-4o model.

| Method | Visual Search (Acc.) | Counting (Acc.) | Scene Desc. (Edit Dist.) | Spatial Rel. (Acc.) |
|---|---|---|---|---|
| Baseline | 48.00 | 16.85 | 1.84 | 43.00 |
| Imaginary-Baseline | 68.00 | 18.24 | 1.75 | 44.00 |
| VISER | **73.00** | **31.55** | **1.63** | **52.50** |

## A.4 Reasoning trace evaluation

To evaluate the impact of visual structure on step-by-step reasoning, we compare two prompting strategies for the 2D scene description task. In the first setting, we provide GPT-4o with the original

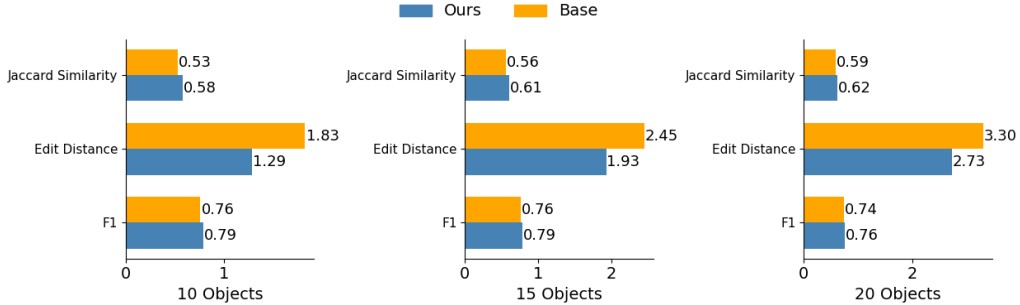

Figure 7: Comparison between VISER and GPT-4o in the 2D scene description task, where models sequentially scan the image row by row to identify objects. Results are reported for scenes with 10, 15, and 20 objects using three evaluation metrics: F1 score, Jaccard similarity, and Edit distance.

image *without any row markings* and use a prompt that instructs: *Divide the image into 4 equal horizontal sections from top to bottom, and list the shape and color of each object in each row.* In the second setting, we explicitly modify the image to include *four horizontal lines* that visually segment the scene into rows. As shown in Figure 7, this structural addition significantly improves performance across all three evaluation metrics (F1 score, Jaccard similarity, and Edit distance) on scenes with 10, 15, and 20 objects. These results suggest that textual instructions alone are insufficient: incorporating structural cues directly into the visual input helps the model follow the intended sequential scanning process more accurately. For consistency, all objects in the scenes were rendered as *colored squares*, and in cases where a square overlapped multiple rows, it was assigned to the row containing the majority of its area. This assignment policy was also reflected explicitly in the prompt.

## B    Benchmark and score details

### B.1    Benchmarks

In this study, we introduce a series of benchmarks designed to evaluate the performance of LVLMs on various visual reasoning tasks, including visual search, counting, scene description, and spatial reasoning. Each benchmark was specifically developed to provide rigorous and systematic assessments of different VLM capabilities, utilizing both synthetic and real-world data. These benchmarks are structured to facilitate controlled experiments while maintaining diversity in task difficulty, scene configurations, and object arrangements. Below, we provide detailed descriptions of each benchmark used in our evaluation.

#### B.1.1    Visual Search Benchmark

Building on the same synthetic generation framework from [9], we adapted the pipeline with task-specific configurations for visual search evaluation.

2D Scenes: We created scenes containing 20, 30, 40, and 50 objects. Each configuration comprised 100 images (50 with targets present and 50 absent), totaling 400 2D scenes.

3D Scenes: Using the same object count progression, we generated 50 images per count (25 with targets present and 25 absent), resulting in 200 3D scenes. The reduced quantity accounts for the higher rendering complexity.

Natural image datasets are not suitable for this evaluation, as they lack two critical properties: (1) the ability to systematically adjust distractor characteristics (shape, color, and spatial arrangement) and (2) precise control over target-distractor similarity levels. Our synthetic paradigm provides this essential controllability, allowing a rigorous assessment of binding capabilities through progressive difficulty scaling while maintaining a balanced presence/absence of the target (50% distribution).

### B.1.2 Counting Benchmark

To assess the counting ability of LVLMs, we developed a comprehensive benchmark based on the generation methodology outlined in [9].

2D Scenes: We generated synthetic 2D scenes with object counts ranging from 10 to 20, incremented in steps of 2 (10, 12, 14, 16, 18, 20). For each object count, we produced 100 distinct images, resulting in a total of 600 2D images.

3D Scenes: Similarly, we created 3D scenes with object counts following the same progression (10 to 20, step size = 2). We generated 50 images per object count, yielding a total of 300 3D images.

To assess real-world generalizability, we incorporated the "Learning to Count Everything" benchmark [32]. We used approximately 300 images containing up to 20 objects to maintain reasonable task difficulty.

This structured approach enables a rigorous assessment of counting performance in controlled synthetic and real-world scenes.

### B.1.3 Scene Description Benchmark

Following [9], we define a *feature triplet* as any set of three objects where one pair shares a feature (e.g., color) and another pair shares a different feature (e.g., shape). For example: green X, green triangle, yellow triangle forms a triplet through shared green and triangle features.

To evaluate scene description capabilities in LVLMs, we developed a comprehensive benchmark using the feature triplet methodology from [9] with controlled object counts and systematically varied triplet configurations.

2D Scenes: We generated synthetic 2D scenes with 10, 15, and 20 objects. For 10 objects, we created configurations with 5 to 20 triplets in increments of 5 (5, 10, 15, 20 triplets). For 15 objects, we extended the triplet counts up to 50 (in steps of 5), and for 20 objects up to 70 triplets (also in steps of 5), with 50 images generated for each configuration.

3D Scenes: Following the same progression, we created 3D scenes with identical objects and triplet counts, generating 30 images per triplet count to account for rendering complexity.

The increasing number of feature triplets introduces progressively greater interference among object features, making the binding problem more challenging and testing the model's reasoning capabilities under increasingly difficult conditions.

This approach enables systematic assessment of scene description performance across: (1) controlled object counts, (2) graduated levels of feature interference through triplet counts, and (3) both 2D and 3D representations. The detailed results for 10, 15, and 20 objects and triplet configurations are presented in Figures 8, 9, and 10, respectively.

### B.1.4 Spatial Relationship Benchmark

To evaluate spatial reasoning in LVLMs, we developed a comprehensive benchmark using the generation methodology from [9] complemented by natural images.

2D Scenes: We generated 200 synthetic 2D scenes with controlled object configurations. For each scene, we created multiple-choice questions testing spatial relationships (top-left, top-right, bottom-left, bottom-right).

3D Scenes: Similarly, we produced 200 synthetic 3D scenes with three-dimensional arrangements, following the same question generation protocol as the 2D scenes.

For real-world evaluation, we used 200 natural images from the benchmark [38], which provides diverse object arrangements in unconstrained environments.

This approach enables systematic assessment of spatial reasoning across: (1) fundamental 2D relationships, (2) complex 3D configurations, and (3) real-world scenarios.

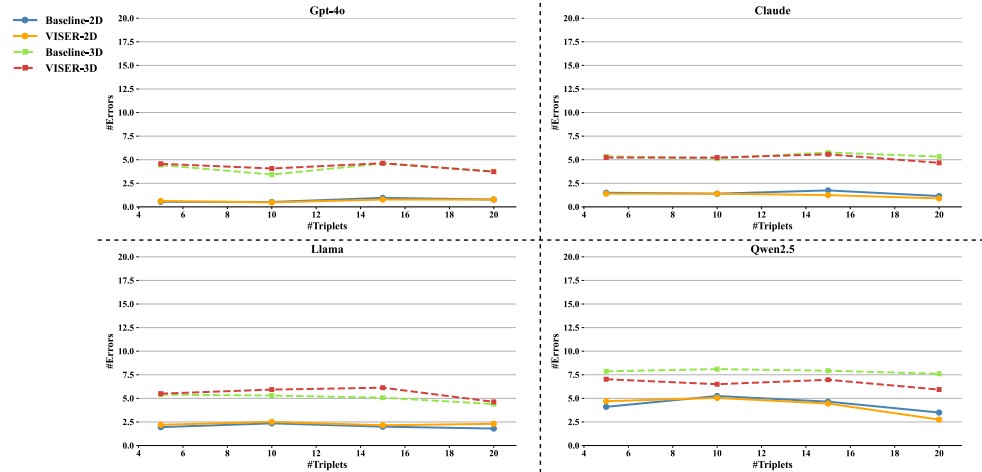

Figure 8: Results of the 10 objects Scene Description task across varying numbers of triplets for 2D and 3D scenes, using different VLM models (GPT-4o, Claude-Sonnet, Qwen2.5-VL, and LLaMa4).

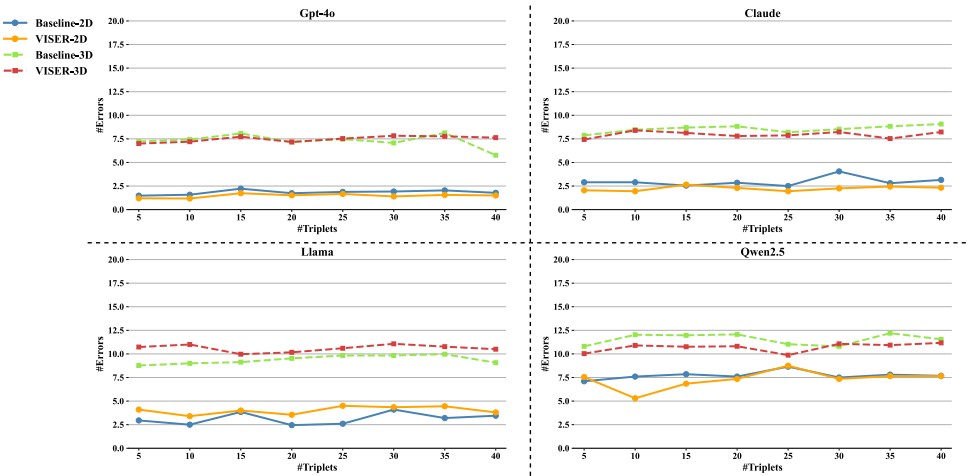

Figure 9: Results of the 15 objects Scene Description task across varying numbers of triplets for 2D and 3D scenes, using different VLM models (GPT-4o, Claude-Sonnet, Qwen2.5-VL, and LLaMa4).

## B.2    Score Metrics

To evaluate the performance of different models across visual reasoning tasks, we employ five metrics: Edit Distance, Accuracy, F1 score, Jaccard Similarity, and Mean Squared Error (MSE). Each metric is applied according to the nature of the task, and formal definitions are provided below.

### 1. Edit Distance (Scene Description)

For the scene description task (both 2D and 3D), we define a custom edit distance that penalizes discrepancies between the predicted and ground truth object lists. The metric is computed in the following steps:

1. **Exact Matches**: All objects with matching *shape* and *color* are removed from both sets.
2. **Partial Matches**:
   - Objects with the same shape but different colors incur a penalty of 1.

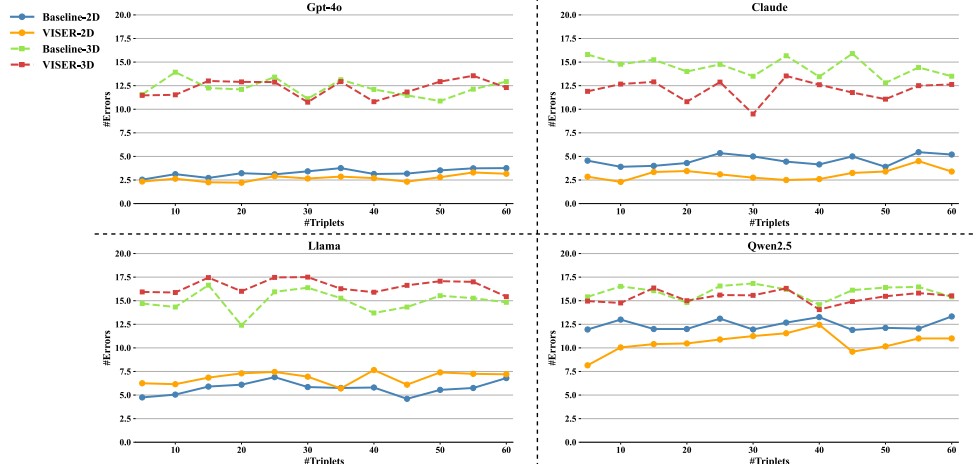

Figure 10: Results of the 20 objects Scene Description task across varying numbers of triplets for 2D and 3D scenes, using different VLM models (GPT-4o, Claude-Sonnet, Qwen2.5-VL, and LLaMa4).

- Objects with the same color but different shapes incur a penalty of 1.

3. **Missed Ground Truth Objects**: Any remaining unmatched ground truth objects are penalized with 2 points.

The total edit distance is calculated as:

$$\text{EditDistance} = 1 \times N_{\text{partial}} + 2 \times N_{\text{missed}} \tag{2}$$

where $N_{\text{partial}}$ is the number of shape-only or color-only matches, and $N_{\text{missed}}$ is the count of completely missed objects. This asymmetric formulation reflects the observation that models more often miss objects than hallucinate them.

## 2. Accuracy (Visual Search, Counting, Spatial Relationship)

Accuracy is used for classification-based tasks and is defined as the ratio of correct predictions to total predictions:

$$\text{Accuracy} = \frac{N_{\text{correct}}}{N_{\text{total}}} \tag{3}$$

This metric is used in visual search, counting and spatial relationship evaluations.

## 3. F1 Score (Reasoning Trace Evaluation)

F1 score measures the harmonic mean between precision and recall and is particularly useful for object-level evaluation:

$$\text{F1} = 2 \cdot \frac{\text{Precision} \cdot \text{Recall}}{\text{Precision} + \text{Recall}} \tag{4}$$

with:

$$\text{Precision} = \frac{TP}{TP + FP}, \quad \text{Recall} = \frac{TP}{TP + FN} \tag{5}$$

where TP (true positives) are objects correctly predicted with matching shape and color, FP (false positives) are incorrect predictions, and FN (false negatives) are missed objects.

## 4. Jaccard Similarity (Reasoning Trace Evaluation)

Jaccard similarity measures the set-level overlap between prediction and ground truth:

$$\text{Jaccard} = \frac{|P \cap G|}{|P \cup G|} \tag{6}$$

where $P$ and $G$ are the sets of predicted and ground truth objects respectively, and equality requires both shape and color to match.

Table 9: Mean squared error (MSE) of counting task across GPT-4o, Claude3.5-sonnet, LLaMa4, and Qwen2.5-VL on 2D, 3D and natural image datasets using base models and VISER.

| | GPT-4o | | Claude-sonnet | | LLaMa4 | | Qwen2.5-VL | |
|---|---|---|---|---|---|---|---|---|
| | Baseline | VISER | Baseline | VISER | Baseline | VISER | Baseline | VISER |
| 2D | 7.50 | **1.33** | 5.32 | **5.25** | **5.515** | 5.575 | 15.03 | **3.89** |
| 3D | 6.13 | **2.44** | **2.28** | 3.01 | 2.74 | **1.53** | 22.04 | **6.21** |
| Natural | 37.33 | **6.33** | 1048.63 | **42.20** | 6.30 | **5.89** | 35.10 | **24.32** |

## 5. Mean Squared Error (Counting)

For the counting task, we compute the mean squared error (MSE) between the predicted and true object counts:

$$\text{MSE} = \frac{1}{N} \sum_{i=1}^{N} (\hat{y}_i - y_i)^2 \tag{7}$$

where $\hat{y}_i$ is the predicted count and $y_i$ is the ground truth count for scene $i$, and $N$ is the total number of scenes. MSE penalizes large deviations in count predictions more heavily.

**MSE Results**: Table 9 presents the Mean Squared Error (MSE) for object counting across the evaluated models. While our primary analysis focused on accuracy metrics, the MSE results provide complementary insights, particularly regarding the magnitude of counting errors.

VISER demonstrates consistent error reduction compared to baseline performance. The most notable improvements occur in natural image scenarios, where exact count prediction remains challenging but our approach achieves substantial MSE reduction (Claude3.5-sonnet: 42.20 versus 1048.63). Similar improvements are evident in synthetic environments, with GPT-4o (1.33 versus 7.50 in 2D) and Qwen2.5-VL (3.89 versus 15.03 in 2D; 6.21 versus 22.04 in 3D) showing significant decreases in squared error.

These MSE results complement our accuracy findings by demonstrating that VISER not only increases correct predictions but also reduces the severity of counting errors when they occur. The notable MSE improvement on natural images, despite the inherent difficulty of exact counting in real-world scenes, highlights VISER's ability to prevent large counting deviations.

### B.3 Performance Variance and Statistical Significance

Each reported score in our experiments is averaged over a substantial number of samples (typically 50100 per configuration), which provides a stable estimate of performance. To minimize non-determinism, all evaluations use greedy decoding (temperature = 0), ensuring deterministic outputs for each input. This reduces variance introduced by stochastic generation and makes the results reproducible. While we do not repeat each setup multiple times, we emphasize robustness by evaluating across a broad set of conditions, including four tasks, both 2D and 3D synthetic data, real-world datasets, and multiple open- and closed-source models.

Across 120 pairwise comparisons between VISER and the baseline (excluding supplementary material), VISER outperforms in 103 cases, shows decreases in 17 cases, and yields ties in 7 cases. Most drops occur with LLaMA4, a model that generally performs poorly on these tasks. Considering only non-tied cases (96 wins vs. 17 losses), a one-sided binomial sign test yields a p-value of $7.4 \times 10^{-15}$, strongly rejecting the null hypothesis that VISER is no better than the baseline. The tied cases typically correspond to trivial scenarios with zero or saturated performance.

## C Implementation Details

In this section, we describe the prompts used across our four evaluation tasks. While we maintain a consistent prompt template for all models, each task requires a tailored instantiation to address its specific objectives. In Figures 11, 12, 13, and 14, we detail the exact prompt formulations employed for each task, which produce the results reported in the Experiments section. Additionally, to assess the models Chain-of-Thought (CoT) reasoning ability, we prepend Lets think step by step

to each prompt, encouraging the model to articulate intermediate reasoning before arriving at its final prediction.

**Baseline**

**Task:**

You are presented with an image containing a set of letters, specifically the letters 'L' and 'T'. These letters will appear in either red or green. Your task is to determine if there are any green 'L's in the image. Follow these steps carefully:
1. Describe each shape in the image, noting their color.
2. Conclude your response by stating [True] if the letter 'L' appears in green, or [False] if there are no green 'L's.

Enclose your final answer in square brackets (Final Answer: []), as shown.

**VISER**

**Task:**

You are presented with an image containing a set of letters, specifically the letters 'L' and 'T'. These letters will appear in either red or green. Your task is to determine if there are any green 'L's in the image. Follow these steps carefully:
1. Describe each shape in the image, noting their color.
2. Conclude your response by stating [True] if the letter 'L' appears in green, or [False] if there are no green 'L's.

Enclose your final answer in square brackets (Final Answer: []), as shown.
Scan sequentially based on horizontal lines exist in the image.

Figure 11: Displaying the prompt used for the 2D scenes in the visual search task. The only difference between the two prompts is the additional instruction: Scan sequentially based on horizontal lines exist in the image.

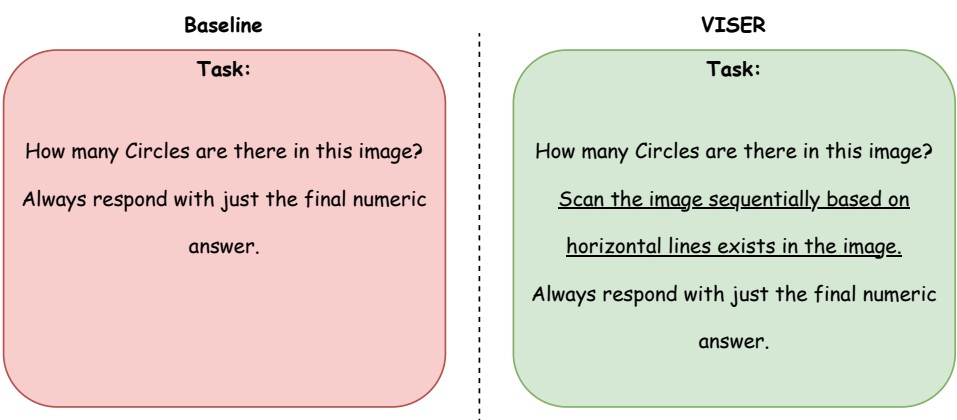

**Baseline**

**Task:**

How many Circles are there in this image?

Always respond with just the final numeric

answer.

**VISER**

**Task:**

How many Circles are there in this image?

Scan the image sequentially based on

horizontal lines exists in the image.

Always respond with just the final numeric

answer.

Figure 12: Displaying the prompt used for the 2D scenes in the counting task. The phrase Scan sequentially based on horizontal lines exist in the image is added to our prompts, in contrast to the baseline input.

## D  Input and Output examples

In this section, we provide a series of examples to demonstrate the outputs of our row-wise structure across different tasks, including visual search, counting, scene description, and spatial relationship analysis. Each task is illustrated with both synthetic (2D and 3D) and, where applicable, real-world scenes. For each task, we compare the outputs of applying VISER with those of a baseline model, highlighting the impact of our approach on a VLM.

Figure 13: Showing the prompt used in our 2D scene description task. The phrase Scan sequentially based on horizontal lines exist in the image is the only addition to our prompt.

Figure 14: Showing the prompt that we use in the 2D scenes for the spatial relationship task. The only difference from the baseline is the added instruction: The grid lines are added to help you compare the objects better.

## D.1 Visual Search

For the visual search task, we present input and output examples from both 2D and 3D datasets (shown in Figures 15 and 16, respectively). In the first example (Figure 15), the objective was to locate the green 'L' among a set of 'L' and 'T' letters within the scene. The baseline model failed to detect the target, returning a [False] output. In contrast, applying row-wise structure successfully identified the green 'L' and returned a [True] output, demonstrating its effectiveness. This pattern was similarly observed in the 3D scene, where the baseline model again failed to locate the target object, while applying VISER accurately detected the presence of the described object within the scene.

## D.2 Counting

For the counting task, in addition to the 2D and 3D scene examples, we incorporate a sample from the natural-image dataset utilized in the evaluation of Learning to Count Everything [32]. The corresponding examples are presented in Figures 17, 18, and 19, respectively. In all three cases, the VLM model misses some objects or hallucinates extra objects and inaccurately counts the number of objects. In contrast, when our proposed method is applied, the VLM model provides the correct object count.

## D.3 Scene Description

In Figures 20 and 21, we present two synthetic examples (one from a 2D scene and the other from a 3D scene) both relevant to the scene description task. The VLM initially fails to accurately identify all objects and their respective attributes within the scenes. However, by applying our proposed method, we improve the model's performance, enabling it to more accurately and precisely describe the objects in the scenes.

## D.4 Spatial Relationship

For the spatial relationship task, we present three examples: one from a 2D synthetic scene (Figure 22), one from a 3D synthetic scene (Figure 23), and one from the natural Spatial Reasoning dataset [38] (Figure 24). In all of these examples, the VLM initially selects the wrong choice. However, adding grid lines helps the model better compare the spatial relationships between the objects, leading to more accurate answers.

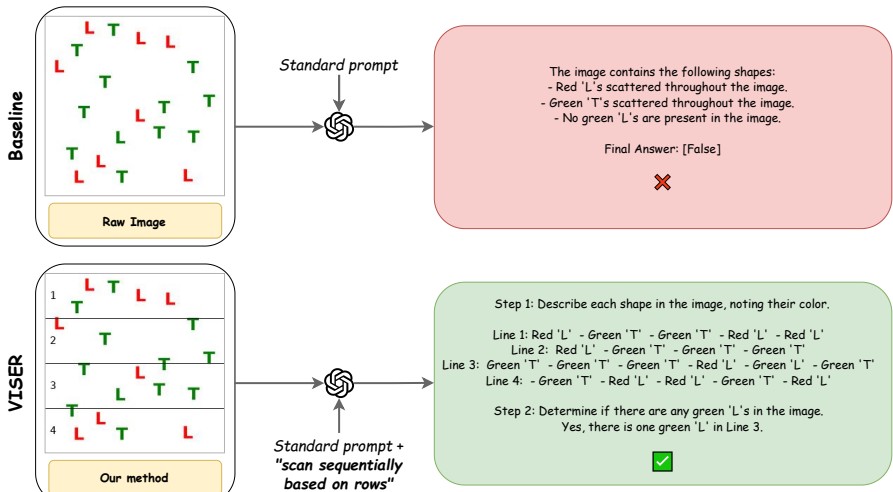

Figure 15: An example of a 2D visual search task illustrating the impact of our proposed method on the output of the VLM model. The baseline fails to detect the green 'L', returning [False], while our row-wise structure successfully identifies it.

## E Failure case

VISER adds lines in a fixed configuration, independent of image content. While this design ensures consistency and simplicity, it can introduce limitations. For instance, added lines may intersect with important objects, potentially introducing ambiguity about those objects. Alternatively, objects may lie in regions where the added lines provide little to no benefit in decomposing the task. To address this, we conduct two additional controlled experiments on the 2D counting task, targeting conditions where our intervention may help or hurt.

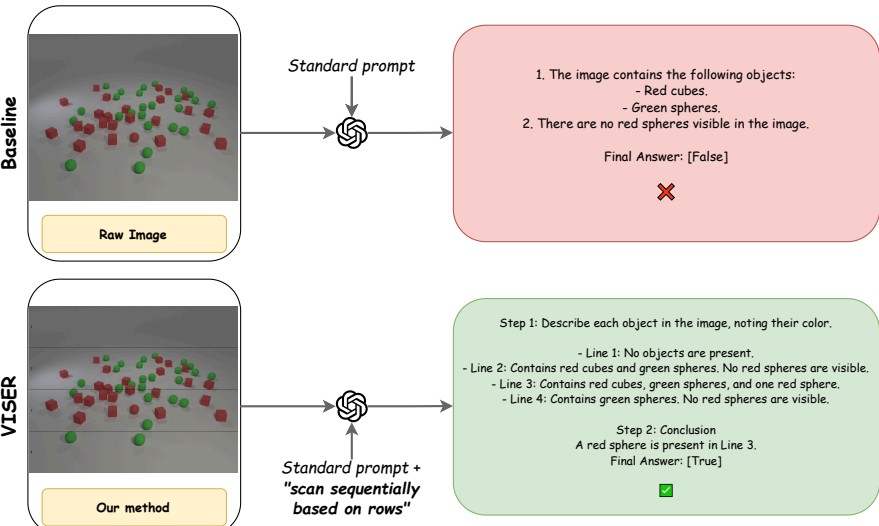

Figure 16: An example of a 3D visual search task illustrating the impact of our row-wise structure on the VLM model's output. The baseline fails to detect the red sphere, returning [False], while applying VISER successfully detects it and returns [True].

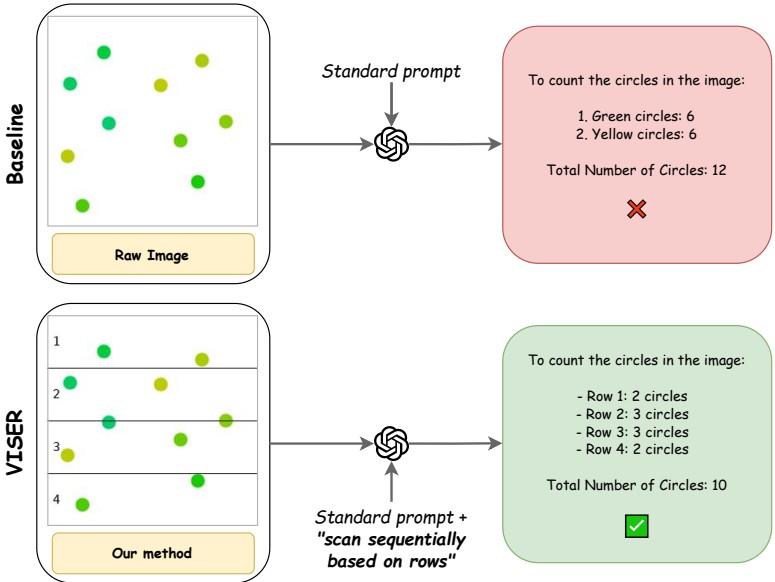

Figure 17: An example of a 2D counting task and the impact of our proposed method on the output of the VLM model. The baseline approach incorrectly counts the number of circles in the 2D scene, whereas our row-wise structure method enables the VLM to accurately count the number of circles.

## E.1 Object-Line Interaction Analysis

We synthetically vary the level of object-line interference in scenes with circular objects and group data into bins (0.0–0.2: low interference; 0.8–1.0: high interference). As illustrated in Table 10, Our method improves performance significantly in low-interference cases. However, in high-interference scenarios (e.g., dense line-object overlaps), the benefit diminishes. This highlights a key failure mode: when scaffolding introduce ambiguity in object detection. For example, overlapping lines can cause objects to be either double-counted (across both lines) or missed entirely (not clearly belonging to either region), leading to reduced model accuracy. Interestingly, GPT-4os baseline

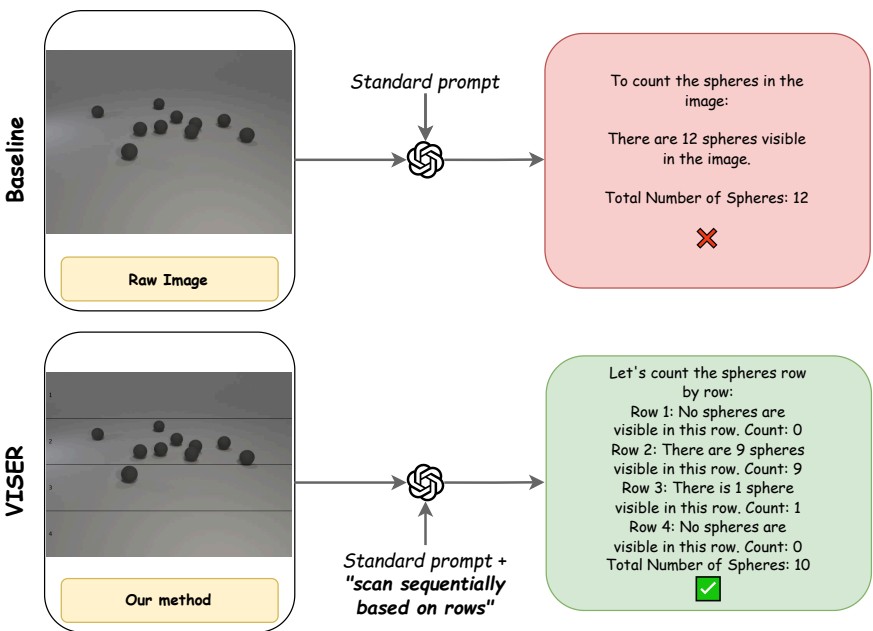

Figure 18: An example of a 3D counting task and the impact of our proposed method on the VLM model's output. The baseline method inaccurately counts the number of spheres in the 3D scene, whereas applying VISER enables the VLM to accurately count the number of spheres.

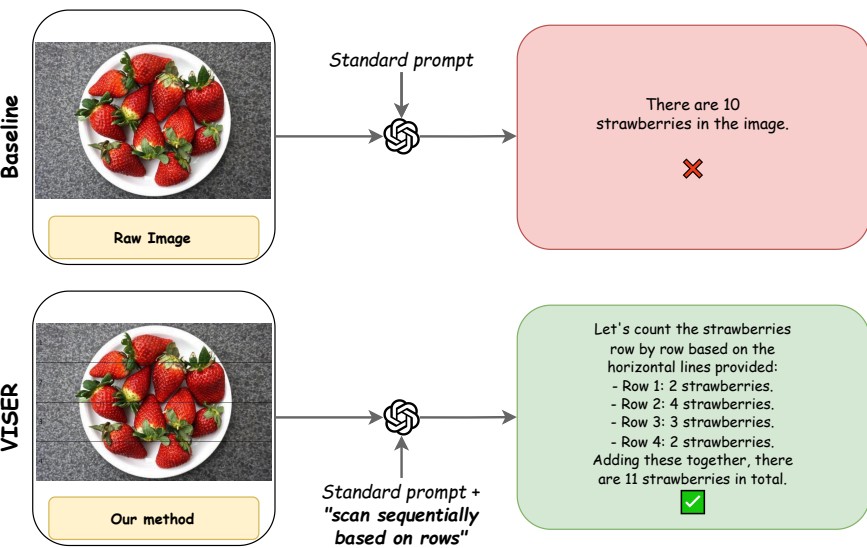

Figure 19: An example of a natural dataset for the counting task and the impact of our proposed method on the output of the VLM model. The baseline method fails to accurately count the number of strawberries in the dataset, whereas row-wise structure enhances the VLM's ability to determine the strawberry count correctly.

accuracy drops with increased density, while Qwen2.5-VL shows the opposite trend. An example of this synthetic dataset is illustrated in Figure 25a.

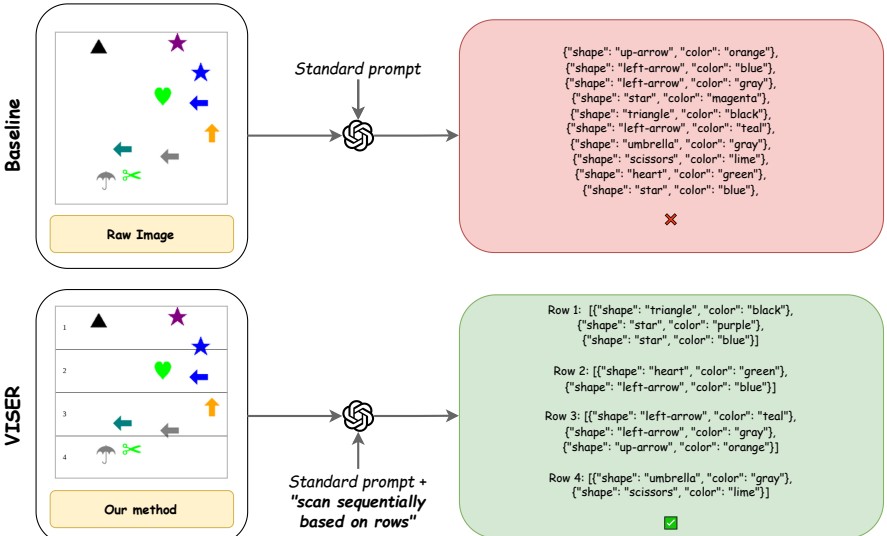

Figure 20: An example of a 2D scene description task and the impact of VISER on the VLM's output. The VLM fails to describe the purple star shape in the scene, whereas VISER correctly describes all the objects.

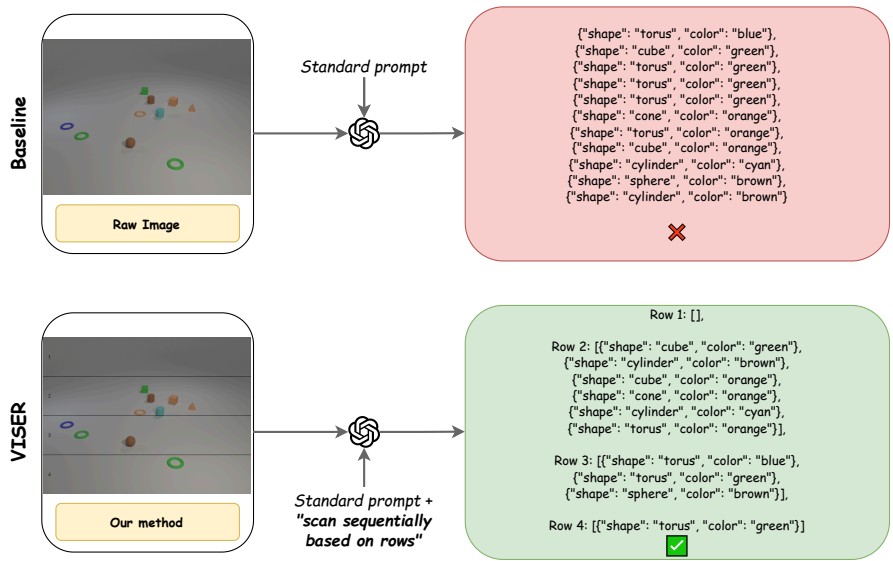

Figure 21: An example of a 3D scene description task and the improvement gained by applying VISER. The VLM incorrectly adds an extra object, counting three green toruses instead of two, which aren't part of the scene. In contrast, our proposed method accurately describes only the objects present.

## E.2 Object Distribution (Entropy) Analysis

To probe another failure case, we conducted a controlled analysis targeting such potential failure modes. Specifically, we evaluated performance with respect to object distributions, which quantifies how widely objects are spread across rows. Low entropy indicates objects are concentrated in one row, while high entropy corresponds to a uniform spread. As illustrated in Table 11, VISER is more effective when objects are distributed across the image (high entropy), where scaffolding pro-

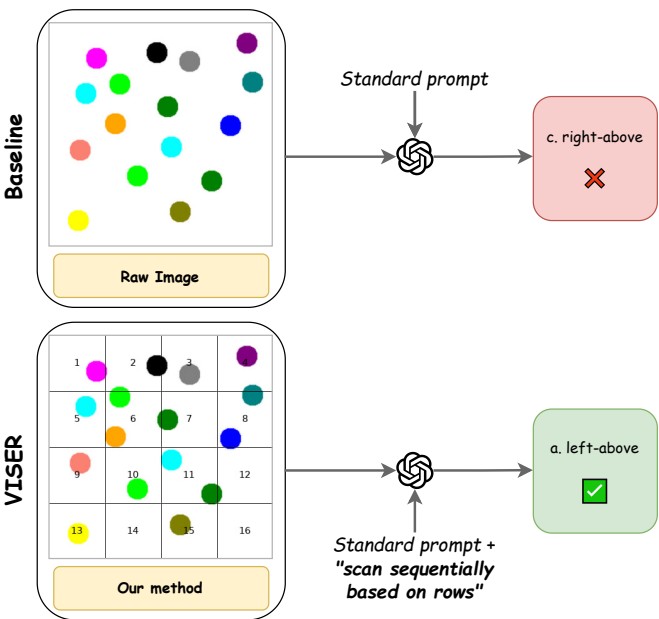

Figure 22: An example of a 2D spatial relationship task and the effect of applying our propsed method on the VLMs output. As shown in the figure, adding grid lines assists the VLM in selecting the correct choice.

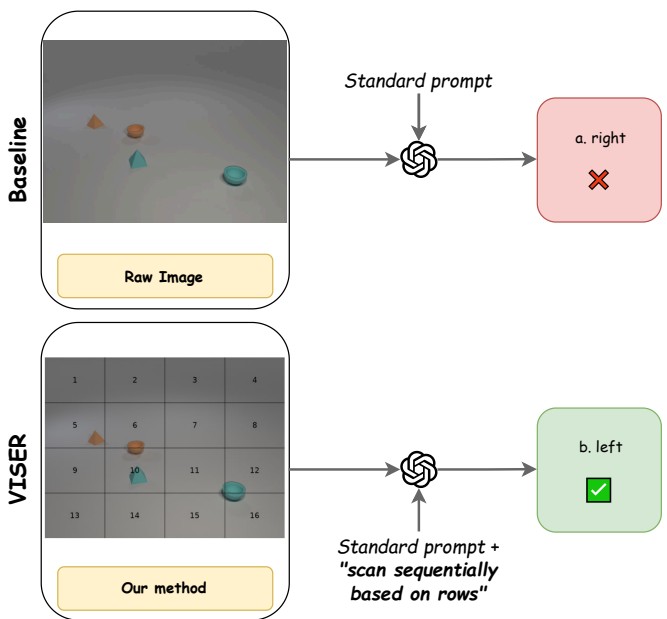

Figure 23: An example of a 3D spatial relationship task and the impact of applying VISER on the VLMs output. As shown in the figure, adding grid lines improves the VLMs ability to select the correct choice.

vides meaningful visual separation. However, when most objects are located in a single region (low entropy), scaffolds are less helpful, sometimes even harmful. This exposes a second failure case:

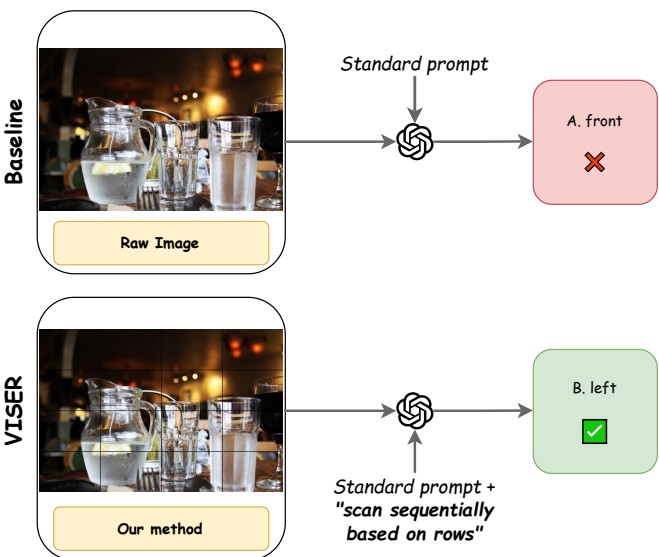

Figure 24: An example from a natural dataset for the spatial relationship task and the impact of applying VISER on the VLMs output. The question asks about the spatial relationship between the jar and glasses from the cameras perspective. As shown in the figure, applying VISER helps the VLM answer the question correctly.

Table 10: Accuracy of GPT-4o and Qwen2.5-VL on the counting task across different levels of average line-object intersection (ranging from 0.00.2 to 0.81.0).

| Model | Method | 0.0-0.2 | 0.2-0.4 | 0.4-0.6 | 0.6-0.8 | 0.8-1.0 |
|---|---|---|---|---|---|---|
| GPT-4o | Baseline | 9.50 | 12.00 | 7.00 | 3.50 | 3.50 |
| | VISER | **26.00** | **26.00** | **22.00** | **14.00** | **13.00** |
| Qwen2.5-VL | Baseline | 3.00 | 4.00 | 10.50 | **13.50** | **18.00** |
| | VISER | **37.00** | **34.50** | **17.50** | 5.00 | 9.00 |

ineffective intervention when object positioning limits scaffold utility. An example of this synthetic dataset is illustrated in Figure 25b.

Table 11: Accuracy of GPT-4o and Qwen2.5-VL on the counting task across different levels of object spatial entropy (ranging from 0.000.50 to 1.752.00).

| Model | Method | 0.00-0.50 | 0.50-1.00 | 1.00-1.25 | 1.25-1.50 | 1.50-1.75 | 1.75-2.00 |
|---|---|---|---|---|---|---|---|
| GPT-4o | Baseline | 29.00 | 30.00 | 27.00 | 33.00 | 38.00 | 37.00 |
| | VISER | 48.00 | 54.00 | 62.00 | 62.00 | 69.00 | 68.00 |
| Qwen2.5-VL | Baseline | 14.00 | 16.00 | 17.00 | 16.00 | 7.00 | 5.00 |
| | VISER | 1.00 | 15.00 | 16.00 | 26.00 | 33.00 | 36.00 |

Our primary experiments are conducted on randomly generated datasets with naturally high entropy ( 1.9) and low average interference ( 0.174), where our method is most effective.

## F  Attention visualization

In this section, we analyze and visualize the attention maps corresponding to the generated outputs from both the baseline and our proposed method using the Qwen2.5-VL-7B-Instruct [41] model.

Figure 26 shows the results of VISER, where we instruct the model to "Scan sequentially based on horizontal lines exist in the image," encouraging it to analyze each row independently. The

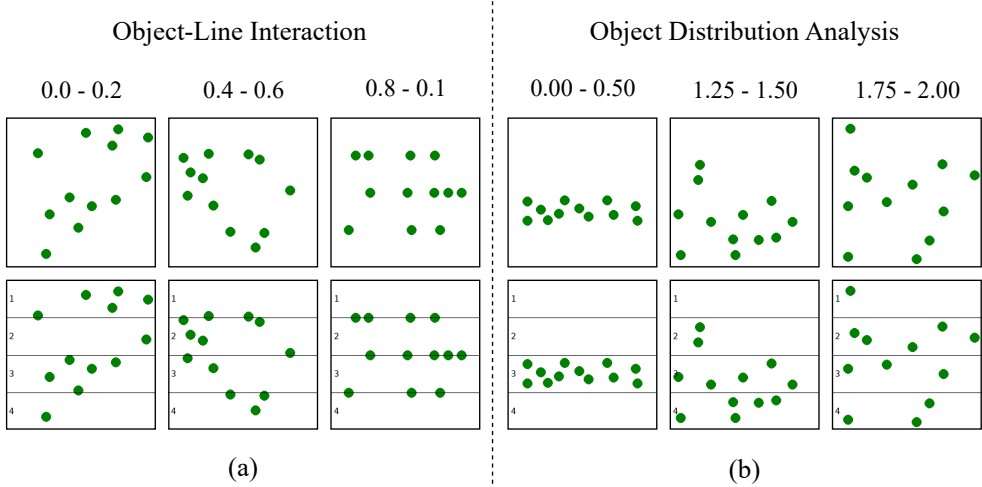

| Object-Line Interaction | | | Object Distribution Analysis | | |
|---|---|---|---|---|---|
| 0.0 - 0.2 | 0.4 - 0.6 | 0.8 - 0.1 | 0.00 - 0.50 | 1.25 - 1.50 | 1.75 - 2.00 |

(a)                                               (b)

Figure 25: Visualization of failure cases in the 2D counting task. Subfigure (a) shows examples with varying levels of object-line interference, while Subfigure (b) illustrates different object spatial distributions.

extracted attention maps for each predicted numerical value confirm this approach, demonstrating that the model attends more strongly to the corresponding spatial regions. For comparison, Figure 27 presents the results from the baseline model, which was given the same instruction. In this case, the attention is less focused, and the predicted numbers do not align clearly with their respective regions. This shows that adding the horizontal line guidance enhanced the model's ability to focus on each region independently, resulting in improved performance and greater precision.

By applying VISER, the model is guided to focus sequentially on smaller, more localized regions within the image. This enables it to more accurately identify and attend to objects in specific areas. The localized approach allows for a more detailed and focused analysis, leading to better alignment between the predicted numerical values and the corresponding spatial regions. In contrast, the baseline method attempts to attend to the entire image in a single step, without such localized guidance. This broader attention scope reduces the model's ability to focus on specific regions, causing difficulty with precise object identification and reducing accuracy. The model fails to effectively resolve finer details and spatial relationships within the image. To further illustrate this, Figure 28 shows the baselines predicted numbers, which attempt to attend to all circles simultaneously. Due to the high number of circles, the model fails to answer the question correctly. Conversely, Figure 26 demonstrates that by incorporating explicit sequential reasoning, VISER allows the model to count objects in each region more accurately and aggregate the results more effectively.

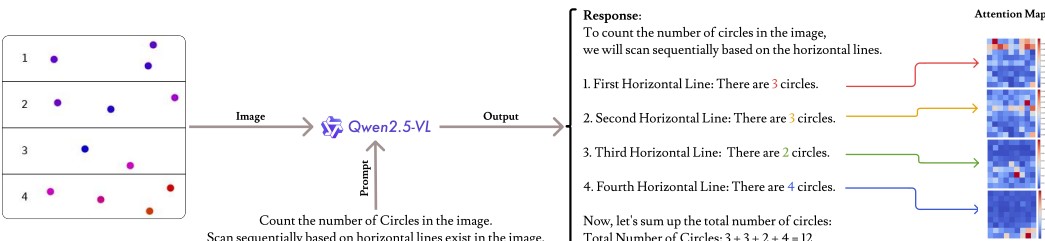

Figure 26: Visualization of 2D attention maps generated by our proposed method using the Qwen2.5-VL-7B-Instruct model on a counting task. The attention patterns demonstrate that the predicted numerical values for each row attend more strongly to their corresponding spatial regions, indicating improved alignment between predictions and visual content.

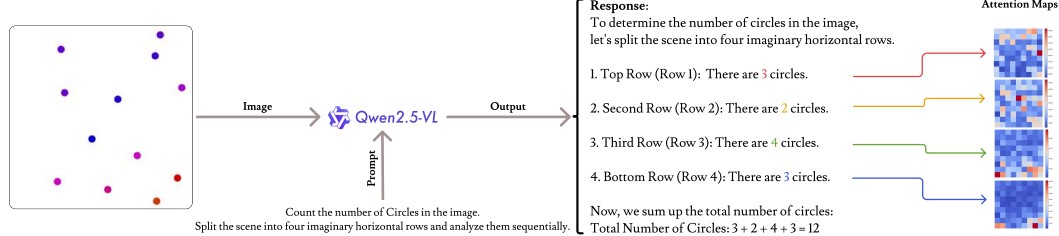

Figure 27: Visualization of 2D attention maps generated by the baseline method using the Qwen2.5-VL-7B-Instruct model on the same counting task. Although provided with the same instruction to scan the image row by row, the attention maps exhibit less focus, and the predicted numerical values show weaker alignment with their corresponding spatial regions. Additionally, the numerical predictions lack proper row-region binding, and the predictions for the third and fourth rows are incorrect.

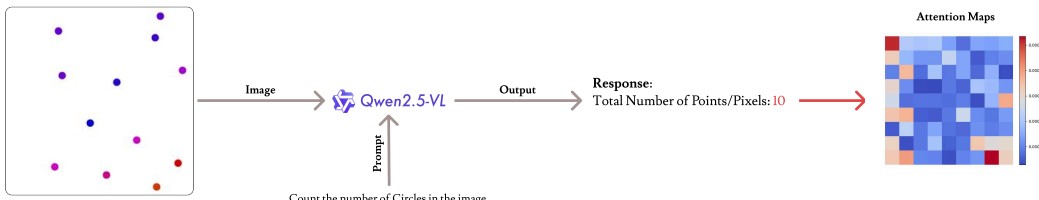

Figure 28: Visualization of the 2D attention maps generated by the baseline method using the Qwen2.5-VL-7B-Instruct model on a counting task. The attention maps reveal a broader, less focused attention pattern, with the model attempting to attend to all regions simultaneously. This lack of localized attention results in incorrect predictions, as evidenced by the model's failure to properly align with the specific spatial regions in the image, leading to errors in counting.

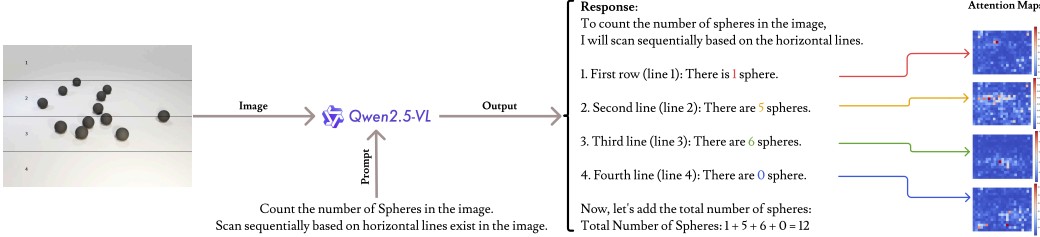

Figure 29: Visualization of 3D attention maps generated by our proposed method using the Qwen2.5-VL-7B-Instruct model on a counting task. The attention patterns demonstrate that the predicted numerical values for each row attend more strongly to their corresponding spatial regions, indicating improved alignment between predictions and visual content.

# G  Computational cost

To ensure a fair comparison of computational cost, we measured the average number of output tokens generated across different tasks. Specifically, we compared three methods using the GPT-4o model: the Baseline, our proposed method (VISER), and a Chain-of-Thought (CoT) prompting strategy. Table 12 reports the average token count for each method across four task types, each evaluated on 100 samples.

Table 12: Comparison of average generated tokens for VISER, CoT, and the Baseline across different tasks using the GPT-4o model.

| Method | Visual Search | Counting | Scene Desc. | Spatial Rel. |
|---|---|---|---|---|
| Baseline | 83.88 | 11.83 | 61.77 | 8.43 |
| CoT | 90.48 | 103.12 | 62.13 | 12.52 |
| VISER | 153.39 | 40.94 | 62.67 | 10.50 |

