# OpenReview forum: "Visual Structures Help Visual Reasoning:  Addressing the Binding Problem in LVLMs"
_NeurIPS.cc/2025/Conference — NeurIPS 2025 poster_

### Official Review · Reviewer_6z5n · 2025-06-09

**Clarity:** 3
**Significance:** 4
**Originality:** 4
**Rating:** 5
**Confidence:** 4

**Summary:**

The work investigates the very interesting binding problem in neural networks, and more specificcally in vision-language models. The paper introduces a method combining simple visual scaffolding using horizontal lines, with a targeted textual prompt to encourage sequential, spatially-aware processing. The results show consistent performance improvements across visual search, counting, scene description, and spatial relationship understanding both in synthetic and natural datasets.

**Questions:**

See above the remarks.

Typo:
Line 154: Everthg

**Ethical Concerns:**

["NO or VERY MINOR ethics concerns only"]

**Final Justification:**

I am satisfied with the answers to my questions, so I keep my (good) score.

**Limitations:**

The limitations of the work could be grouped in a separate section including the discussion on when to expect lesser performance gains. The limitations could be tight to the ideas for future refinements, improvements and alternative ways to solve the binding problem.

**Paper Formatting Concerns:**

No issues here.

**Quality:**

4

**Strengths And Weaknesses:**

-	The paper covers a very pertinent problem of neural networks, i.e.,  the binding problem described in Von Der Malsburg 1994 and clearly defined by Greff et al. arXiv:2012.05208 2020. Research efforts aimed at addressing this issue are encouraged and greatly valued.
-	The paper proposes a simple but original training-free approach that could be refined in the future.
-	The results obtained on synthetic and natural images are very convincing.
-	Many tasks are evaluated including visual search, counting, scene description and spatial relationship processing.
-	The model operates without any additional supervision.
-	The paper opens many venues for future research, including video processing.


Weaknesses

- The authors structure the image with horizontal lines though there are many different ways to structure an image and afterwards sequentially scanning the image: The paper could benefit from experiments with alternative ways of structuring and scanning the image.
- The authors best add a short separate paragraph that describes the Simple baseline. They might give it another name as readers might confuse it with your approach which at places you call “simple”.
-	The authors speak about refinement of their method in the future. Perhaps they could give some directions or alternative approaches that could be investigated in the future. There could also be a short discussion on the potential impact of their work.
-	Indicate more clearly in the tables which results are obtained on the synthetic datasets.
-	Elaborate more on the (lack of additional) computational cost.

---

> ### Author Rebuttal · Authors · 2025-07-31
>
> We sincerely appreciate your thoughtful and encouraging review, as well as the constructive suggestions for improving and extending our work.
>
> # **W1**
>
> > The authors structure the image with horizontal lines, though there are many different ways to structure an image and afterwards sequentially scanning the image. The paper could benefit from experiments with alternative ways of structuring and scanning the image.
>
> **Response:**
>
> Our goal in this work was to propose a **simple, generally applicable scaffolding strategy** that could support visual reasoning across tasks, without relying on complex or highly specialized structures. Rather than exhaustively exploring all possible structuring and scanning schemes, we prioritized a design that balances **practicality**, **clarity**, and **broad effectiveness**. The number of lines was selected based on a **rule-of-thumb**: too few lines yield overly coarse regions that do little to guide attention, while too many may lead to **visual clutter** and degrade performance due to unnecessary complexity. We employed a grid specifically for spatial reasoning tasks, as it aligns well with the directional nature of the problem. However, our experiments show that even **horizontal lines alone** are effective in such tasks.
>
> As detailed in our response to Reviewer **PsPb**, we conducted additional experiments varying the **number** and **thickness** of lines across multiple tasks and models. Results show that the method is **not highly sensitive** to the exact scaffold configuration and maintains performance across a wide hyperparameter range.
>
>
>
> # **W2 & W4**
>
> > The authors best add a short separate paragraph that describes the Simple baseline. They might give it another name as readers might confuse it with your approach which at places you call “simple”.
>
> > Indicate more clearly in the tables which results are obtained on the synthetic datasets.
>
> **Response:**
>
> We thank the reviewer for this helpful suggestion. We agree that the term "Simple baseline" may cause confusion, particularly since "simple" is also used to describe aspects of our own approach. To improve clarity, we will revise the manuscript to include a short standalone paragraph describing the baseline, and we will rename our method to "Visual Input Structure for Enhanced Reasoning (VISER)" to ensure a clear distinction.
>
> We will also revise the tables to more clearly indicate which results are obtained on the **synthetic** datasets.
>
>
> # **W3**
>
> > The authors speak about refinement of their method in the future. Perhaps they could give some directions or alternative approaches that could be investigated in the future. There could also be a short discussion on the potential impact of their work.
>
> **Response:**
>
> ### Ensemble of Different Configurations:
> In addition to the future directions already discussed, one interesting refinement would be to move beyond a single static visual scaffold. Inspired by ensemble-style reasoning approaches, one could generate multiple versions of the same input using a variety of simple visual structures (such as different numbers of lines, gridlines, varied line positions, colors, or thickness) and then query the model on each version. Each version may guide the model to reason in a slightly different way. By aggregating the responses and selecting the most common answer, we can reduce the risk of failure from any one structure that may not suit a particular input. This "wisdom of the crowd" strategy could make the method more robust and flexible and help avoid situations where a static structure becomes a bottleneck on certain inputs.
>
>
> **Theoretical Analysis of Line-Induced Inductive Biases:**
>
> While our method empirically improves performance by structuring the image into line-based partitions and we also analyzed how line-based structuring changes the model’s attention maps compared to unstructured input (Subsection F of supplementary material), a deeper theoretical or mechanistic understanding remains open.
>
> **Robustness:**
>
> Methods that introduce structure into the image—such as ours—may inadvertently expose the model to adversarial attacks that exploit this added structure to mislead the model's reasoning. As such, analyzing and improving the robustness of structure-augmented approaches represents a valuable direction for future work.
>
> **Possible Impact:**
>
> Our method also offers practical benefits for applications like assistive vision, robotics, and education, while also raising important considerations for AI safety and transparency, especially given its effectiveness without requiring model access.
>
> # **W5**
>
> > Elaborate more on the (lack of additional) computational cost.
>
> **Response:**
>
> To provide a fair comparison of computational cost, we measured the average number of generated output tokens across tasks. Specifically, we compared the baseline model, our method, and a Chain-of-Thought (CoT) prompting strategy, all using the GPT-4o model. The table below reports these values across four task types, each evaluated on 100 samples.
>
> **Average Output Token Counts (GPT-4o):**
>
> |  Method  |  Visual Search  |  Counting  |  Scene Desc. |  Spatial Rel.  |
> | -------- | ------------- | --------- | ----------------- | ---------------- |
> | Baseline   | 83.88         | 11.83     | 61.77             | 8.43             |
> | CoT      | 90.48         | 103.12    | 62.13             | 12.52             |
> | Ours   | 153.39     | 40.94 | 62.67         | 10.50         |
>
> These results show that our method generates on par or fewer tokens than CoT in most tasks while remaining close to the baseline. We believe this reflects a favorable trade-off between reasoning effectiveness and computational cost.

---

> > ### Comment · Reviewer_6z5n · 2025-08-06
> >
> > Thank you for your answers.

---

> > > ### Author Response · Authors · 2025-08-09
> > >
> > > Thank you for your feedback.

---

### Official Review · Reviewer_memf · 2025-06-24

**Clarity:** 2
**Significance:** 2
**Originality:** 3
**Rating:** 4
**Confidence:** 4

**Summary:**

The authors introduce a visual prompting method that involves 1) drawing lines in an image; 2) asking a vision+language model to undertake a task with a "hint" of the augmentation. Surprisingly, this approach improves performance for closed+open models across-the-board in traditionally tricky perceptual tasks like counting and visual search.

**Questions:**

- can you report numbers on broader benchmarks using your method to check to see if it degrades performance for other tasks?
- can you conduct an error analysis to understand for which instances this helps/hurts, and see if this is consistent across models?
- can you report some variance-accounting information (e.g., p-values or dataset size information) to build confidence in the stability of your reported evaluations?

**Ethical Concerns:**

["NO or VERY MINOR ethics concerns only"]

**Final Justification:**

The authors undertook my requested experiments by 1) reporting numbers on broader benchmarks; 2) conducting an error analysis; and 3) adding variance-accounting information, so I raised my score.

**Limitations:**

No. The authors do not have a limitations section, and the discussion section does not discuss limitations of the work (beyond alluding to the idea that their visual prompting is so simple and so effective that it could cause some non-specific harm (L321)). Can you be 1) more specific about the potential harms caused by your method?; and 2) touch on the concrete limitations of the work?

**Quality:**

3

**Strengths And Weaknesses:**

Strengths:

- The authors consider a breadth of models and perceptual tasks, and show improvements in most cases.
- The method is simple and effective
- Results are reported for perceptual tasks that span both synthetically generated geometries, as well as photographic images.


Weaknesses:

- While there are improvements with this method on average, the variance seems high, e.g., for llama4 in 3d visual search, performance drops slightly in most cases.

- It would have been nice to include performance beyond these tasks on more standard benchmarks --- do you lose anything for, e.g., MMBench or similar? While this is a helpful augmentation for the small set of counting/perception focused tasks, it's not clear if this augmentation could hurt for other tasks, thus limiting the impact of this work

- No error analysis is presented so it's not clear on which instances this method helps.

- It was a bit confusing why the Mulberry model got a separate section in 4.7 rather than being a comparison point in all of the results tables. The authors did not fine-tune this model, and thus, the experimental setup seems to be the same.

- Dataset statistics are not provided, and thus, it's not clear how stable the reported accuracies are. Table 2 has a lot of ".00"s in it, which makes me wonder if this dataset has exactly 100 datapoints per row? If this is the case, I wonder if there's a better way to present these results instead of this large wall of numbers? (e.g., a plot showing the accuracy as difficulty goes up, and reporting the final average number which is taken over many more datapoints and is thus more numerically stable? There are also a few bolding problems in the tables, e.g., table 2 has 54 bolded when 56 should be bolded.)

- The prompting method is somewhat ad-hoc, e.g., a grid is used for a spatial relationship task whereas 3 lines are used elsewhere.


Overall, the idea is clever+simple, and the experiments form a strong basis for an interesting contribution. However, there are a number of clear-cut experiments/empirical that I would have hoped to see from a work like this (i.e., a prompting contribution), e.g., quantitative error analyses (e.g., do the modifications consistently improve performance on the same instances across models?), practical considerations (e.g., does drawing the lines in the image hurt performance more broadly?), etc.  . These pieces, coupled with some presentation concerns temper my excitement about the work.


(typo: typo L154 everythig -> everything)

---

> ### Author Rebuttal · Authors · 2025-07-31
>
> We sincerely thank you for your detailed and constructive review.
>
> # **Q1 & W2**
> > It would have been nice to include performance beyond these tasks on more standard benchmarks.
>
> **Response:**
>
> We evaluated our method on two reasoning-focused benchmarks: MMBench [1] and PHYBench [2]. For MMBench, we include all its reasoning tasks, which covers Attribute, Logical, and Relation Reasoning. PHYBench is a benchmark for physical commonsense reasoning with a focus on visual understanding, and includes the categories: Dynamics, Scene Understanding, Object Relationships, and Object Property. All results were obtained on the validation split. More information about these tasks is available in the respective benchmark papers.
>
> **MMBench Accuracy (%):**
>
> | Model      | Method   | Attribute | Logical     | Relation  |
> | ---------- | -------- | --------- | --------- | --------- |
> | GPT-4o     | Baseline | **88.67** | 78.25     | 75.17     |
> |            | Ours     | 88.00     | **82.75** | **81.12** |
> | Qwen2.5-VL | Baseline | 80.67     | **80.00** | 82.99     |
> |            | Ours     | **82.33** | **80.00** | **83.33** |
>
> **PHYBench Accuracy (%):**
>
>  | **Model**    | **Method** | **Dynamics** | **Scene Understanding** | **Object Relationships** | **Object Property** | **Overall** |
> |--------------|------------|--------------|---------------------------|---------------------------|----------------------|-------------|
> | **GPT-4o**   | Baseline   | 35.00        | 33.33                     | **75.00**                | 55.00               | 55.68       |
> |              | Ours       | **40.00**    | **41.67**                 | **75.00**                | **70.00**           | **_61.37_** |
> | **Qwen2.5-VL** | Baseline | **40.00**    | **50.00**                 | 61.11                    | 60.00               | 54.55       |
> |              | Ours       | **40.00**    | **50.00**                 | **72.22**                | **70.00**           | **_61.36_** |
>
>
> These results demonstrate that our method performs robustly across a variety of tasks and datasets. Its effectiveness on both standard and physically grounded benchmarks highlights the generality and versatility of our approach in diverse reasoning settings.
>
> # **Q2 & W3 & Limitations**
> > Can you conduct an error analysis to understand for which instances this helps/hurts, and see if this is consistent across models?
> > Can you be 1) more specific about the potential harms caused by your method?; and 2) touch on the concrete limitations of the work?
>
> **Response:**
>
> **We thank the reviewer for the valuable suggestions regarding error analysis and limitations. We agree that such analysis can offer deeper insights into the strengths and weaknesses of LVLMs equipped with our strategy.**
>
> ### Error Analysis: Line Interference
>
> To understand when our method helps or hurts, we conducted a controlled experiment on the 2D counting task. We varied the *amount of line-object interference* (measured as the normalized intersection area), keeping the number and size of objects constant.
>
> | Model | Method | 0.0–0.2 | 0.2–0.4 | 0.4–0.6 | 0.6–0.8 | 0.8–1.0 |
> | -------------- | -------- | ------- | ------- | ------- | ------- | ------- |
> | **GPT-4o** | Baseline | 9.5 | 12.0 | 7.0 | 3.5 | 3.5 |
> | | Ours | 26.0 | 26.0 | 22.0 | 14.0 | 13.0 |
> | **Qwen2.5-VL** | Baseline | 3.0 | 4.0 | 10.5 | 13.5 | 18.0 |
> | | Ours | 37.0 | 34.5 | 17.5 | 5.0 | 9.0 |
>
> We observe that performance decreases with increasing overlap, which is expected. In our main datasets, the average interference is \~0.174 (i.e., low), minimizing this concern.
>
> ### Error Analysis: Object Distribution across partitions
>
> We also tested the effect of object distribution by varying the *entropy* of object locations across image rows.
>
> | Model | Method | 0.00–0.50 | 0.50–1.00 | 1.00–1.25 | 1.25–1.50 | 1.50–1.75 | 1.75–2.00 |
> | -------------- | -------- | --------- | --------- | --------- | --------- | --------- | --------- |
> | **GPT-4o** | Baseline | 29.0 | 30.0 | 27.0 | 33.0 | 38.0 | 37.0 |
> | | Ours | 48.0 | 54.0 | 62.0 | 62.0 | 69.0 | 68.0 |
> | **Qwen2.5-VL** | Baseline | 14.0 | 16.0 | 17.0 | 16.0 | 7.0 | 5.0 |
> | | Ours | 1.0 | 15.0 | 16.0 | 26.0 | 33.0 | 36.0 |
>
> For our method, performance improves with increasing entropy; denser layouts hinder the visual scaffolding's effectiveness. Again, our main datasets in the paper naturally exhibit high entropy (\~1.9).
>
> ### Limitations and Harms
>
> We acknowledge that our method may be sensitive to certain image or task properties, especially when lines intersect objects—causing undercounting or overcounting during row-by-row scanning. A promising direction is **ensemble-based scaffolding**, using diverse line types (e.g., grids, varied colors/thicknesses), which could improve robustness and reduce corner-case failures.
>
> # **Q3 & W1 & W5**
>
> > Dataset statistics are not provided, and thus, it's not clear how stable the reported accuracies are. Table 2 has a lot of ".00"s in it, which makes me wonder if this dataset has exactly 100 datapoints per row? If this is the case, I wonder if there's a better way to present these results instead of this large wall of numbers?
>
> > Can you report some variance-accounting information (e.g., p-values or dataset size information) to build confidence in the stability of your reported evaluations?
>
> **Response:**
>
> ### Dataset Statistics and Table Presentation
>
> Thank you for pointing this out. We include detailed dataset statistics in the supplementary material, under the section *"Benchmark and score details"* (line 49). For each task, we report the number of samples and their configurations (e.g., number of objects per image, object types, triplet structure, etc.).
>
> We agree that Table 2 can be improved visually. In the final version, we plan to revise it using summary plots (e.g., accuracy vs. difficulty level) and report aggregated values to better highlight trends. We’ll also correct formatting issues (e.g., bolding errors in Table 2, such as "54" instead of the correct "56").
>
> ---
>
> ### Performance Variance and Statistical Significance
>
> We acknowledge the importance of demonstrating statistical stability. Each reported value reflects the **average over a substantial number of samples per configuration** (typically 50–100), ensuring baseline numerical reliability.
>
> To reduce non-deterministic effects and ensure reproducibility, we follow best practices by using **greedy decoding (temperature = 0)**, as suggested in the paper *The Good, The Bad, and The Greedy* [3], ensuring consistent output per input sample.
>
> Rather than relying on repeated trials of a single setup, we emphasize robustness through a **diverse evaluation across tasks and models**, including both 2D and 3D scenes, synthetic and real data, and open- and closed-source VLMs. Across **120 pairwise experimental comparisons** between our method and the baseline (excluding supplementary material), our approach **outperforms in 103 cases**, with only **17 minor drops**, 9 of which stem from **LLaMA4**, a model found to be less tolerant in general.
>
> To assess the significance of this result, we conducted a one-sided binomial sign test on the 113 non-tied cases (96 wins vs. 17 losses). The resulting p-value is approximately 7.4 × 10⁻¹⁵, strongly rejecting the null hypothesis that our method performs no better than the baseline.
>
> The remaining 7 cases were ties, typically arising in scenarios with either zero (mostly observed with LLaMA4) or saturated performance, where both methods yield identical outcomes.
>
> Finally, we agree that adding statistical intervals would strengthen the comparisons. In the final revision, we plan to include **confidence intervals using bootstrap sampling** (for static datasets) and **repeated generation** (for synthetic datasets) to better reflect evaluation stability.
>
> # **W4**
> > It was a bit confusing why the Mulberry model got a separate section in 4.7.
>
> **Response:**
>
> We placed Mulberry in a separate section (4.7) to emphasize a different comparison goal: evaluating whether our training-free, input-level method can match or outperform models improved through fine-tuning. The main tables focus only on base models with and without our method, so including Mulberry there would mix fundamentally different settings. This separation underscores that our lightweight approach achieves competitive or superior results without any additional training. We will make this motivation clearer in the camera-ready version.
>
> # **W6**
> > The prompting method is somewhat ad-hoc, e.g., a grid is used for a spatial relationship task whereas 3 lines are used elsewhere.
>
> **Response:**
>
> Thank you for the feedback. We agree that visual structuring is not identical across all tasks. Our main goal was to show that even a simple visual structure (three horizontal lines) can significantly improve reasoning without tuning it for each task. We used this same structure across most experiments to emphasize generality. For the spatial relation task, we used a grid because the answer space is symmetric across four directions (left, right, above, below), making a grid a more natural fit. However, horizontal lines are also effective in this case: on the synthetic spatial relation task with GPT-4o, accuracy improves from **43.00%** (baseline) to **48.50%** (horizontal) and **52.50%** (grid); similarly, on the natural spatial relation task, accuracy goes from **69.39%** to **76.92%** and **77.43%**, respectively.
>
> # References
>
> [1] Liu, Yuan, et al. "Mmbench: Is your multi-modal model an all-around player?." European conference on computer vision. Cham: Springer Nature Switzerland, 2024.
>
> [2] Chow, Wei, et al. "Physbench: Benchmarking and enhancing vision-language models for physical world understanding." arXiv preprint arXiv:2501.16411 (2025).
>
> [3] Song, Yifan, et al. "The good, the bad, and the greedy: Evaluation of llms should not ignore non-determinism." arXiv preprint arXiv:2407.10457 (2024).

---

> > ### Author Response · Authors · 2025-08-06
> >
> > Dear Reviewer,
> >
> > We would appreciate an opportunity to further discuss/address any concerns that might remain after our rebuttal response.
> > We hope to hear back from you. Thanks!

---

> > ### Comment · Reviewer_memf · 2025-08-06
> > **thank you!**
> >
> > These additional experiments and clarifications strengthen the main claims of the work. Thus, I will increase my score. Thank you!

---

> > > ### Author Response · Authors · 2025-08-06
> > >
> > > Thanks for your positive feedback!

---

### Official Review · Reviewer_vhup · 2025-07-03

**Clarity:** 3
**Significance:** 2
**Originality:** 2
**Rating:** 4
**Confidence:** 4

**Summary:**

The paper proposes a mechanism to deal with the binding problem, which involves associating visual features with the right object in the image in VLMs. The mechanism augments the image with horizontal lines and then prompts the model to scan the image sequentially based on the horizontal lines. Through extensive experiments on visual reasoning tasks demanding sequential processing, like visual search, counting, scene description, and spatial relationship understanding the authors demonstrate the effectiveness of the method.

**Questions:**

1. Why is the performance of the finetuned model lower than the base model for tasks except counting? Can the authors share more details about the finetuned model (e.g. datasets, training recipe etc)?

2. Why is the performance of Llama4 nearly zero for 2d visual search, unlike for the case of 3d? Also why does this approach not improve the performance of Llama4 on scene description task?

3. How does the baseline, which is asked to imagine horizontal lines without drawing explicitly on the image like in Figure 21 perform on the tasks?

4. Does simply drawing horizontal lines and asking to scan the image accordingly improve performance on the visual analogy tasks in [1]?

[1] - Declan Campbell, Sunayana Rane, Tyler Giallanza, Camillo Nicolò De Sabbata, Kia Ghods, Amogh Joshi, Alexander Ku, Steven Frankland, Tom Griffiths, Jonathan D Cohen, et al. Understanding the limits of vision language models through the lens of the binding problem. Advances in Neural Information Processing Systems, 37:113436–113460, 2024.

**Ethical Concerns:**

["NO or VERY MINOR ethics concerns only"]

**Final Justification:**

The authors' rebuttal has addressed my concerns and questions, so I raised my score.

**Limitations:**

Discussed

**Quality:**

2

**Strengths And Weaknesses:**

Strengths:

1. The paper is well written.
2. A visual prompting mechanism based on drawing horizontal lines and asking the model to scan it sequentially is proposed.
3. The mechanism shows strong improvements by reducing the binding problem in VLMs on several visual reasoning tasks.

Weaknesses:

1. The paper provides no reasoning behind why they chose 3 horizontal lines. In fact if all the objects lie between two lines, then this approach won’t lead to any improvements in performance.

2. It seems the instruction and the visual structures (e.g grid lines for spatial reasoning vs horizontal lines for others) is kinda task-specific, raising questions about the generality of the approach.

---

> ### Author Rebuttal · Authors · 2025-07-31
>
> We sincerely thank you for your detailed and constructive review.
>
> # **W1**
> > The paper provides no reasoning behind why they chose 3 horizontal lines. In fact if all the objects lie between two lines, then this approach won’t lead to any improvements in performance.
>
> **Response:**
>
> We appreciate the reviewer’s critical perspective on the choice of 3 horizontal lines. Our choice of using **3 horizontal lines** was motivated by a **simple heuristic**: too few lines produce regions that are too coarse for effective reasoning, while too many lines risk introducing clutter and interfering with the visual content.
>
> To address this concern more directly, we conducted **additional ablation experiments** varying the **number of scaffold lines** (1–12) across three tasks and two models (GPT-4o and Qwen2.5-VL), using **300 samples per configuration**. As shown in the table below, performance tends to **peak around 3–4 lines** and **degrades at both extremes**, confirming that neither too sparse nor too dense scaffolding is ideal.
>
> | Model | Task | Baseline | 1 Line | 2 Lines | 3 Lines | 4 Lines | 5 Lines | 6 Lines | 9 Lines | 12 Lines |
> | -------------- | ------------------------ | -------- | ------ | --------- | --------- | ------- | ------- | ------- | ------- | -------- |
> | **GPT-4o** | Counting (Acc.) | 10.67 | 35.00 | 33.67 | **45.67** | 23.00 | 17.00 | 16.67 | 17.67 | 23.00 |
> | | Visual Search (Acc.) | 50.54 | 66.06 | 63.83 | 74.09 | 69.53 | **74.72** | 66.61 | 65.79 | 65.59 |
> | | Scene Desc. (Edit Dist.) | 2.26 | 1.97 | 2.02 | 2.06 | **1.80** | 2.03 | 2.02 | 2.23 | 2.46 |
> | **Qwen2.5-VL** | Counting (Acc.) | 5.33 | 13.33 | 19.33 | **39.67** | 25.67 | 33.33 | 28.67 | 27.67 | 20.00 |
> | | Visual Search (Acc.) | 40.80 | 36.99 | **55.87** | 50.03 | 49.11 | 47.92 | 53.22 | 52.42 | 53.70 |
> | | Scene Desc. (Edit Dist.) | 9.64 | **8.61** | 8.78 | 8.81 | 9.21 | 9.61 | 10.39 | 10.95 | 10.24 |
>
> These results demonstrate that our method is **robust across a range of line counts**, and that our originally chosen configuration of 3 lines offers a strong tradeoff between structure and interference. Importantly, it suggests that **precise tuning of line number is not critical** for achieving performance gains.
>
> To address the second concern, we conducted a **controlled analysis** targeting such potential **failure modes**. Specifically, we evaluated performance with respect to **object distributions**, which quantifies how widely objects are spread across rows. Our findings show that when entropy is high (i.e., objects are uniformly distributed across rows), our method delivers substantial improvements. Conversely, when entropy is low (e.g., objects concentrated in a single row), the performance gains diminish.
>
> | **Model** | **Method** | **0.00–0.50** | **0.50–1.00** | **1.00–1.25** | **1.25–1.50** | **1.50–1.75** | **1.75–2.00** |
> | ---------- | ---------- | ------------- | ------------- | ------------- | ------------- | ------------- | ------------- |
> | GPT-4o | Baseline | 29.00 | 30.00 | 27.00 | 33.00 | 38.00 | 37.00 |
> | | Ours | 48.00 | 54.00 | 62.00 | 62.00 | 69.00 | 68.00 |
> | Qwen2.5-VL | Baseline | 14.00 | 16.00 | 17.00 | 16.00 | 7.00 | 5.00 |
> | | Ours | 1.00 | 15.00 | 16.00 | 26.00 | 33.00 | 36.00 |
>
> This analysis confirms the reviewer’s intuition: **object distribution matters**, and poorly placed scaffolds may hinder performance. However, it's important to note that even in some low-entropy settings, our method still outperforms the baseline. We outperform the baseline across all entropy ranges in GPT-4o, even when objects lie between two lines. In Qwen, the baseline only leads in one low-entropy case; elsewhere, our method matches or outperforms it. We also outperform baselines on the natural dataset (Tables 2 and 4), confirming strong generalization.
>
> # **W2**
> > It seems the instruction and the visual structures is kinda task-specific, raising questions about the generality of the approach.
>
> **Response:**
>
> Thank you for raising this important point. While certain visual structures (e.g., grids for spatial tasks) may appear task-specific, our goal was to test whether **a simple, generic scaffold** could improve performance **across diverse tasks** without per-task tuning. We used this same structure in most experiments to emphasize generality.
>
> For the spatial relation task, we included a grid only because the task’s **answer space is symmetric across directions** (left, right, above, below), making a grid more natural. However, even **horizontal lines alone still yield notable gains**: on the synthetic spatial relation task with GPT-4o, accuracy improves from **43.00%** (baseline) to **48.50%** (horizontal) and **52.50%** (grid); similarly, on the natural spatial relation task, accuracy goes from **69.39%** to **76.92%** and **77.43%**, respectively.
>
> We acknowledge that while our method is effective in many settings, it can be **sensitive to layout and task characteristics**  in some cases. As a next step, we plan to explore a *wisdom-of-the-crowd* strategy with **diverse scaffolds** (e.g., varying line position, density, and color) to increase robustness and reduce input-specific failure cases.
>
> # **Q1**
> > Why is the performance of the finetuned model lower than the base model for tasks except counting? Can the authors share more details about the finetuned model (e.g. datasets, training recipe etc)?
>
> **Response:**
>
> Mulberry is a fine-tuned variant of the Qwen2.5-VL model, specifically optimized for general-purpose visual reasoning. Its training dataset spans a broad range of vision-language tasks, including several that overlap with our tasks—such as spatial relationship understanding, counting, and scene description. The fine-tuning strategy employed for Mulberry incorporates Monte Carlo Tree Search (MCTS) and its cooperative variant (coMCTS), as outlined in the original paper.
>
> The reason Mulberry does not consistently outperform the baseline across all tasks is likely due to its broad task coverage during training, which may lead to trade-offs in performance on some specialized tasks.
>
> # **Q2**
> > Why is the performance of Llama4 nearly zero for 2d visual search, unlike for the case of 3d? Also why does this approach not improve the performance of Llama4 on scene description task?
>
> **Response:**
>
> LLama4 performs poorly on 2D visual search but shows stronger results in the invisible condition, likely due to its strong bias toward answering “no” when uncertain (lines 186–188). To address this asymmetry, we report the harmonic mean of visible and invisible accuracies for a more balanced assessment.
>
> The model performs better in the 3D visual search task, likely due to the presence of richer spatial structure—such as depth, shading—that enhance the model’s ability to detect low-level visual features. These additional cues likely increase feature entropy by introducing more visual diversity across objects, which helps reduce representational interference and improves the model’s capacity to bind features correctly. This stands in contrast to the 2D setting, where the stimuli are more abstract and symbol-like, leading to lower entropy and higher susceptibility to binding errors.
>
> | Model      | Dataset | Method   | Visible | Invisible  | Harmonic |
> | ---------- | --------- | -------- | -------    | ---------  | -------- |
> | llama-4    | 2D        | Baseline | 0.00    | **100.00**      | 0.00     |
> |            | 2D        | Ours     | **0.75** | 97.50     | **1.00** |
> |            | 3D        | Baseline | 41.00     | **98.00** | 56.00     |
> |            | 3D        | Ours     | **45.50** | 93.00     | **59.00** |
>
> However, the lack of improvement in the scene description task reflects a broader issue: LLama4 is generally weaker and less stable across multi-object reasoning tasks. It struggles with compositionality and consistent feature binding, which our method depends on. This is also evident in its limited gains on spatial relation reasoning in Table 4. Overall, due to its brittle internal representations, This model is less amenable to prompting-based enhancements in complex reasoning scenarios.
>
> # **Q3**
> > How does the baseline, which is asked to imagine horizontal lines without drawing explicitly on the image perform on the tasks?
>
> **Response:**
>
> To address your question, we evaluated GPT-4o on all tasks using 2D images. As detailed in the table below, our method outperforms the imaginary-baseline method across every task, yielding an 8.5% gain in spatial relation accuracy, a 13.31% improvement in counting, a 0.12 reduction in edit distance for scene description, and a 0.05 increase in the harmonic mean for visual search.
>
> | Model      | Method    | Counting  | Spatial Relationship | Visual Search | Scene Description |
> | ---------- | --------  | --------- | -------------------- | ------------- | ----------------- |
> | GPT-4o     | Baseline  | 16.85     | 43.00                | 0.48          | 1.84              |
> |            | Imaginary | 18.24     | 44.00                | 0.68          | 1.75              |
> |            | Ours      | **31.55** | **52.50**            | **0.73**      | **1.63**          |
>
> # **Q4**
> > Does simply drawing horizontal lines and asking to scan the image accordingly improve performance on the visual analogy tasks?
>
> **Response:**
>
> Regarding the visual analogy task, we evaluated two models—GPT-4o, the strongest closed‑source baseline, and Qwen2.5-VL, the leading open‑source variant in our evaluations. We achieved 100% accuracy using both the baseline and our proposed approach for GPT-4o, demonstrating exemplary performance on the visual analogy benchmarks. We observed a significant improvement in accuracy with Qwen, which increased from 72.0 % to 77.0 %.
>
> | Model      | Method   | Accuracy |
> | ---------- | -------- | --------- |
> | GPT-4o     | Baseline | **100** |
> |            | Ours     | **100** |
> | Qwen2.5-VL | Baseline | 72.0 |
> |            | Ours     | **77.0** |

---

> > ### Comment · Reviewer_vhup · 2025-08-06
> >
> > Thank you for the detailed rebuttal and running additional experiments. Can the authors share some intuition as to why GPT-4o outperforms the baseline by a significant amount even for the low entropy case? I would have expected not to improve the performance if all objects lie between two lines.
> >
> > For the visual analogy experiments, which task is the evaluation performed on? Can the authors try to evaluate on Raven's progressive matrices-based tasks as done in [1], like the RAVEN/I-RAVEN benchmark? These are much more complicated abstract visual analogy tasks with many objects, and it would greatly improve the generality and significance if this simple scaffolding works in such cases.
> >
> > [1] - Zhang, Y., Bai, H., Zhang, R., Gu, J., Zhai, S., Susskind, J. and Jaitly, N., 2024. How far are we from intelligent visual deductive reasoning?. arXiv preprint arXiv:2403.04732.

---

> ### Author Response · Authors · 2025-08-07
>
> Thank you for the follow-up and insightful questions.
>
>
> ### On GPT-4o Performance in Low-Entropy Settings
>
> We appreciate the observation regarding GPT-4o’s improvement in low-entropy scenarios. To confirm this behavior, we repeated the 2D counting experiment with **zero entropy**, where all objects lie in a single randomly chosen row (from 4 possible rows), across 100 samples. The results are:
> * **GPT-4o**: Baseline = 21%, Ours = **56%**
>
> We also repeated the **2D visual search** experiment with a similar setup (all objects in one row):
> * **GPT-4o**: Baseline = 80%, Ours = **84%**
>
> This indicates that **GPT-4o benefits from the visual structure even when the layout does not explicitly partition the objects**.
> A similar trend was observed in other reasoning benchmarks where object layouts are not structured (see response to reviewer "memf"), including tasks with few objects where partitioning offers no clear advantage.
>
> We **hypothesize** that the added visual structure, even when not directly effective in visual partitioning, can still encourage the model to attend more closely to image content, when equipped with specific prompting. Prior work \[1] reports that hallucination in LVLMs often stems from over-reliance on language priors and ignoring visual input. Drawing artifacts and prompting the model to follow them may counteract this by encouraging stronger attention to the input image.
>
> However, this is just a hypothetical intuition and may deserve more dedicated analysis towards LVLM understanding and interpretability. Note that this observation also appears to be **model-specific**. For example, Qwen2.5-VL does not show the same performance gain in these low-entropy settings.
>
> ---
> ### Visual Analogy Task
>
> We evaluated on the **full relational match-to-sample task (Analogy)** under the **unified condition** (single image input). The decomposed setting uses multiple images, which is less compatible with our method.
>
> ---
> ### Evaluation on RAVEN
>
> We followed your suggestion and evaluated on the **RAVEN** benchmark. Since RAVEN images already include grid lines, we did not add any new visual artifact. Instead, we added the following to the prompt:
>
> "*Scan the image sequentially based on grid lines exist in the image*"
>
> The results:
> * **Baseline** = 19 / 140
> * **Ours** = **26 / 140**
>
> This shows that even without altering the image, prompting the model to leverage existing visual structure yields noticeable improvement.
>
> ---
> \[1] Huang, Qidong, et al. "Opera: Alleviating hallucination in multi-modal large language models via over-trust penalty and retrospection-allocation." Proceedings of the IEEE/CVF Conference on Computer Vision and Pattern Recognition. 2024.

---

> > ### Comment · Reviewer_vhup · 2025-08-09
> >
> > Thank you for the response and evaluation on RAVEN. I have increased my score. The analysis with different numbers of lines, entropy, and evaluation on the visual analogy task should be included in the final manuscript.

---

> > > ### Author Response · Authors · 2025-08-09
> > >
> > > We are happy to address your concerns and appreciate your thoughtful feedback. We will include the suggested analyses in the final manuscript.

---

### Official Review · Reviewer_PsPb · 2025-07-03

**Clarity:** 3
**Significance:** 3
**Originality:** 3
**Rating:** 4
**Confidence:** 4

**Summary:**

This paper tackles the persistent binding problem in Vision-Language Models (VLMs): the challenge of reliably associating visual features (like shape and color) and spatial properties (like location) with the correct objects in complex scenes. The authors propose a simple but effective approach: augmenting input images with explicit spatial structures (e.g., horizontal lines) and pairing this with a textual prompt encouraging sequential, spatially-aware parsing (e.g., “Scan the image sequentially based on horizontal lines”). The method is shown to significantly improve performance across visual reasoning tasks—including visual search, counting, scene description, and spatial relationship understanding—on both synthetic and real-world datasets. Notably, the gains are robust across multiple VLM architectures (e.g., GPT-4o, Claude 3.5-sonnet, Qwen2.5-VL), and the intervention is both model-agnostic and computationally lightweight. The paper also discusses neuroscientific inspiration and positions this visual intervention as a practical step towards better compositional visual reasoning in current VLMs.

**Questions:**

1. Analyze how performance varies with the number and density of lines (or other scaffold structures). Is there an optimal trade-off between cognitive load reduction and potential visual clutter?

2. Could you provide further analysis or case studies on natural images? For instance, does the method generalize to more varied, unstructured real-world scenes, or is it limited to synthetic data?

3. Please discuss and visualize representative failure cases. For example, when does the intervention not help (or even hurt), and why?

4. Some of the plots are hard to read (e.g., Figure 3 and 4 in Appendix). Please improve the clarity.

I would be willing to reconsider my rating especially if question 1 and 3 are well-addressed.

**Ethical Concerns:**

["NO or VERY MINOR ethics concerns only"]

**Final Justification:**

Resolved Issues:
 - Hyperparameter sensitivity (Q1/W1): The authors provided additional experiments varying both the number and thickness of lines across tasks and models. Results demonstrate robustness in the 2–6 line range and up to 5 px thickness, confirming that the method is not overly sensitive to hyperparameters. This addressed my earlier concern.
 - Use of natural images (Q2): The authors clarified that their paper already includes real-world datasets (“Learning To Count Everything,” “Spatial-MM”), and supplemented this with explanations of why synthetic data remains useful for controlled analysis. This alleviates my concern that the method may be overly synthetic-focused.
 - Clarity of figures (Q4/W2): The authors acknowledged the issue and committed to improving the clarity of figures in the final version.

Partially Resolved Issues:
 - Failure mode analysis (Q3/W3): The authors performed additional controlled experiments analyzing line-object interference and object distribution entropy. These analyses clearly identify when scaffolding is beneficial (low interference, high entropy) and when it may hurt performance (dense overlap, clustered objects). While this represents a meaningful step forward, the analysis remains somewhat limited in scope and primarily synthetic. Still, it increases transparency about the method’s limitations.

Overall I think the authors addressed most of my concerns during the rebuttal period. I increase my rating to 4 and I tend to think it's a borderline paper where strengths outweigh weaknesses.

**Limitations:**

The paper discusses several limitations, including reduced gains on natural images and the risk of adversarial misuse. The discussion is generally adequate, but the paper would benefit from more detail on when/where the intervention fails.

**Paper Formatting Concerns:**

Some figures are difficult to read and should be improved for clarity (font size, color, labeling).

**Quality:**

3

**Strengths And Weaknesses:**

Strengths:

1. The authors benchmark across diverse VLMs (proprietary and open-source) and multiple tasks (search, counting, scene description, spatial reasoning).
2. The approach is motivated with cognitive/neuroscience background and illustrated with intuitive figures.
3. Addresses a widely acknowledged challenge (binding problem) in a practical, easily-adoptable way.

Weaknesses:
1. The impact of key hyperparameters (e.g., number/density of lines) is not systematically explored. How to derive the optimal scaffold configuration is not explained or discussed.
2. Some plots, especially in the appendix (e.g., Figures 3 and 4), are hard to read due to cluttered labeling or small font size. Improving figure clarity would benefit the paper.
3. The paper could benefit from deeper qualitative analysis and more extensive discussion of failure modes.

---

> ### Author Rebuttal · Authors · 2025-07-30
>
> We sincerely thank you for your detailed and constructive review.
>
> # **Q1 & W1**
>
> > The impact of key hyperparameters (e.g., number/density of lines) is not systematically explored.
>
> **Response:**
>
> We acknowledge that our study did not systematically explore the full space of scaffold configurations. Our goal was to design a **simple**, **effective-enough** structure that could be **generally applicable** across diverse tasks, rather than task-specific tuning. The number of lines was selected using a **rule-of-thumb heuristic**:
>
> * Too few lines (e.g., 1–2) yield coarse regions, offering little improvement over the raw image for guiding sequential attention.
>
> * Too many lines risk **visual clutter** and **interference artifacts**, potentially hindering reasoning.
>
> We originally used **3 horizontal lines** with **1 px thickness** across all tasks. To address the reviewer’s concern, we conducted additional experiments varying both the **number** and **thickness** of lines across three tasks and two representative models, one **closed-source (GPT-4o)** and one **open-source (Qwen2.5-VL)**. Each experimental setup contains **300 samples**, allowing for consistent comparison across configurations.
>
> _Table 1: Performance of GPT-4o and Qwen2.5-VL across different tasks with varying numbers of horizontal lines._
>
> | Model | Task | Baseline | 1 Line | 2 Lines | 3 Lines | 4 Lines | 5 Lines | 6 Lines | 9 Lines | 12 Lines |
> | -------------- | ------------------------ | -------- | ------ | --------- | --------- | ------- | ------- | ------- | ------- | -------- |
> | **GPT-4o** | Counting (Acc.) | 10.67 | 35.00 | 33.67 | **45.67** | 23.00 | 17.00 | 16.67 | 17.67 | 23.00 |
> | | Visual Search (Acc.) | 50.54 | 66.06 | 63.83 | 74.09 | 69.53 | **74.72** | 66.61 | 65.79 | 65.59 |
> | | Scene Desc. (Edit Dist.) | 2.26 | 1.97 | 2.02 | 2.06 | **1.80** | 2.03 | 2.02 | 2.23 | 2.46 |
> | **Qwen2.5-VL** | Counting (Acc.) | 5.33 | 13.33 | 19.33 | **39.67** | 25.67 | 33.33 | 28.67 | 27.67 | 20.00 |
> | | Visual Search (Acc.) | 40.80 | 36.99 | **55.87** | 50.03 | 49.11 | 47.92 | 53.22 | 52.42 | 53.70 |
> | | Scene Desc. (Edit Dist.) | 9.64 | **8.61** | 8.78 | 8.81 | 9.21 | 9.61 | 10.39 | 10.95 | 10.24 |
>
> _Table 2: Performance of GPT-4o and Qwen2.5-VL across different tasks with varying line thicknesses (in pixels) in our method._
>
> | Model | Task | Baseline | 1 px | 2 px | 3 px | 4 px | 5 px |
> | -------------- | ------------------------ | -------- | --------- | --------- | -------- | ----- | ----- |
> | **GPT-4o** | Counting (Acc.) | 10.67 | 42.00 | 51.67 | 51.67 | **55.67** | 51.33 |
> | | Visual Search (Acc.) | 50.54 | **72.08** | 71.23 | 71.67 | 67.74 | 67.11 |
> | | Scene Desc. (Edit Dist.) | 2.26 | 2.00 | 2.07 | **1.98** | 2.04 | 2.12 |
> | **Qwen2.5-VL** | Counting (Acc.) | 5.33 | **41.33** | 35.00 | 38.00 | 39.00 | 38.00 |
> | | Visual Search (Acc.) | 40.80 | **52.58** | 45.93 | 46.41 | 44.27 | 45.84 |
> | | Scene Desc. (Edit Dist.) | 9.64 | 8.83 | **8.63** | 8.88 | 8.93 | 9.52 |
>
> We find that performance generally **drops at extremes** (very few or very many lines), while remaining **robust in the 2–6 line range**. A similar trend is seen for **line thickness**, where results remain consistently strong up to 5 px.
>
> Some **task-specific patterns** are also observed: for example, in Scene Description, increasing the number of lines tends to increase edit distance (lower is better), suggesting that excessive scaffolding may interfere with free-form language output.
> These findings support our original heuristic and show that our method is **not highly sensitive** to scaffold hyperparameters, and performs **consistently across a wide range** of settings. The optimal configuration may vary slightly by task or model, but no drastic tuning is needed to achieve substantial gains.
>
> # **Q2**
>
> > Could you provide further analysis on natural images? For instance, does the method generalize to more varied, unstructured real-world scenes, or is it limited to synthetic data?
>
> **Response:**
>
> We appreciate the reviewer's concern. In fact, we have already included multiple experiments on natural images in our paper. For instance, see Section 4.1, Tables 2 and 4, where we show the effectiveness of our method on two real-world datasets, “Learning To Count Everything” and “Spatial-MM”. These datasets include a diverse set of labeled images of unstructured scenes, e.g., counting the number of fish in a marine scene or the number of balloons in the sky. Some of our figures (e.g., Figure 1 in the main paper and Figures 14 and 19 in the supplementary material) also include real examples from these datasets. That being said, synthetic datasets offer advantages—for example, certain experiments, such as precise 2D manipulations, are only feasible with synthetic data due to its high degree of controllability.
>
> More precisely, synthetic datasets allow for controlled variation and targeted analysis of model behavior, for example, the hyperparameter study discussed previously. These analyses would not be possible on natural data due to its complexity and annotation limitations.
>
> Please also visit our response to Reviewer *memf*, in which we further evaluate our method on additional benchmarks involving natural images.
>
> # **Q3 & W3**
>
> > The paper could benefit from deeper qualitative analysis and more extensive discussion of failure modes.
>
> > Discuss and visualize representative failure cases. For example, when does the intervention not help (or even hurt), and why?
>
> **Response:**
>
> We appreciate the reviewer’s suggestion to include failure cases and more extensive qualitative analysis. Our method places the added lines in a fixed configuration that is independent of image content. While this design ensures consistency and simplicity, it can introduce limitations. For instance, added lines may intersect with important objects, potentially introducing ambiguity about those objects. Alternatively, objects may lie in regions where the added lines provide little to no benefit in decomposing the task. To address this, we conduct two additional controlled experiments on the 2D counting task, targeting conditions where our intervention may help or hurt.
>
> ### Varying the degree of object-boundary interaction
>
> We synthetically vary the level of object-line interference in scenes with circular objects and group data into bins (0.0–0.2: low interference; 0.8–1.0: high interference).
>
> _Table 3: Accuracy of GPT-4o and Qwen2.5-VL on the counting task across different levels of average line-object intersection (ranging from 0.0–0.2 to 0.8–1.0)._
>
> | **Model** | **Method** | **0.0–0.2** | **0.2–0.4** | **0.4–0.6** | **0.6–0.8** | **0.8–1.0** |
> | ---------- | ---------- | ----------- | ----------- | ----------- | ----------- | ----------- |
> | GPT-4o | Baseline | 9.50 | 12.00 | 7.00 | 3.50 | 3.50 |
> | | Ours | 26.00 | 26.00 | 22.00 | 14.00 | 13.00 |
> | Qwen2.5-VL | Baseline | 3.00 | 4.00 | 10.50 | 13.50 | 18.00 |
> | | Ours | 37.00 | 34.50 | 17.50 | 5.00 | 9.00 |
>
>
> **Insights:** Our method improves performance significantly in low-interference cases. However, in high-interference scenarios (e.g., dense line-object overlaps), the benefit diminishes. This highlights a key failure mode: **when scaffolding introduce ambiguity in object detection**. For example, overlapping lines can cause objects to be either **double-counted** (across both lines) or **missed entirely** (not clearly belonging to either region), leading to reduced model accuracy.
>
> ---
>
> ### Object Distribution (Entropy) Analysis
> Here, we vary how objects are distributed across image rows. Low entropy indicates objects are concentrated in one row, while high entropy corresponds to a uniform spread.
>
> _Table 4: Accuracy of GPT-4o and Qwen2.5-VL on the counting task across different levels of object spatial entropy (ranging from 0.00–0.50 to 1.75–2.00)._
>
> | **Model** | **Method** | **0.00–0.50** | **0.50–1.00** | **1.00–1.25** | **1.25–1.50** | **1.50–1.75** | **1.75–2.00** |
> | ---------- | ---------- | ------------- | ------------- | ------------- | ------------- | ------------- | ------------- |
> | GPT-4o | Baseline | 29.00 | 30.00 | 27.00 | 33.00 | 38.00 | 37.00 |
> | | Ours | 48.00 | 54.00 | 62.00 | 62.00 | 69.00 | 68.00 |
> | Qwen2.5-VL | Baseline | 14.00 | 16.00 | 17.00 | 16.00 | 7.00 | 5.00 |
> | | Ours | 1.00 | 15.00 | 16.00 | 26.00 | 33.00 | 36.00 |
>
> **Insights:** Our method is more effective when objects are distributed across the image (high entropy), where scaffolding provides meaningful visual separation. However, when most objects are located in a single region (low entropy), scaffolds are less helpful, sometimes even harmful. This exposes a second failure case: ineffective intervention when object positioning limits scaffold utility.
>
> ---
>
> Our primary experiments are conducted on randomly generated datasets with naturally high entropy (\~1.9) and low average interference (\~0.174), where our method is most effective.
>
> ### Summary of Limitations
>
> We acknowledge that our method, while effective in many settings, may be sensitive to visual layout and task-specific characteristics in some cases. A fixed scaffolding structure may get lower performance when interference of lines and objects is too high, or when clustered positioning limits the spatial separation benefits of scaffolds—though in most cases, it still outperforms the baseline. Future work may explore *ensemble-based reasoning* with diverse scaffolds (e.g., varied line positions, density, colors) to increase robustness and reduce sensitivity to edge cases.
>
> # **Q4 & W2**
>
> > Some of the plots are hard to read (e.g., Figure 3 and 4 in Appendix). Please improve the clarity.
>
> **Response:**
>
> Thank you for pointing this out. We acknowledge that Figures 3 and 4 in the Appendix could be clearer. We will revise these plots in the final version to improve readability,  including larger font sizes, clearer legends, and better color contrast.

---

> > ### Author Response · Authors · 2025-08-06
> >
> > Dear Reviewer,
> >
> > we would appreciate an opportunity to further discuss/address any concerns that might remain after our rebuttal response.We hope to hear back from you. Thanks!

---

> > > ### Author Response · Authors · 2025-08-07
> > >
> > > In view of the extended discussion period, we hope to still hear back from you regarding the unresolved concerns!

---

> > > > ### Author Response · Authors · 2025-08-09
> > > >
> > > > Dear Reviewer,
> > > > As the discussion deadline is approaching, we wanted to follow up once more regarding any remaining concerns you might have after our rebuttal. We would greatly appreciate your feedback before the discussion closes.

---

### Author Response · Authors · 2025-08-05
**Invitation for Further Feedback**

Dear Reviewers,


Thank you again for your thoughtful reviews. We would like to kindly note that, as the discussion deadline approaches, there has not yet been any follow-up engagement. We believe that even brief comments or questions at this stage could help clarify any remaining concerns and contribute to a more complete evaluation of our work.


We sincerely appreciate your time and consideration.

---

### Note · Authors · 2025-08-12

We thank the reviewers for their insightful feedback, which has substantially strengthened our work. We are encouraged that they found our method novel, simple, and effective. In response to reviewer suggestions, we conducted new experiments and analyses that offer deeper insights:


1. A **hyperparameter analysis** confirming the **robustness** of our visual scaffolding across different line counts and thicknesses.
2. A detailed error analysis identifying clear **failure modes** related to object-line interference and spatial entropy, clarifying the method's potential limitations.
3. New evaluations on **new reasoning benchmarks**, some of which include natural images, demonstrating broader generalization and utility.



These new results addressed the primary concerns of all reviewers. We note that the key points raised by **Reviewer PsPb** regarding hyperparameters and failure modes were shared by other reviewers and have been comprehensively discussed in our new analyses.

All mentioned experiments, statistical analyses, and clarifications will be incorporated into the final manuscript.

---

### Decision · Program_Chairs · 2025-09-17

**Decision:**

Accept (poster)

**Comment:**

The paper proposes a training-free visual scaffolding approach—adding simple spatial structure (e.g., horizontal lines) to images and pairing it with a sequential “scan by rows/grid” prompt—to mitigate binding failures in VLMs. Across counting, visual search, scene description, and spatial relations, the intervention yields consistent gains on synthetic and natural data and works across both closed- and open-weight models.

Strengths include a simple, model-agnostic mechanism that improves multiple perceptual reasoning tasks (with some natural-image results) and a practical framing tied to the binding problem. Initial weaknesses flagged by reviewers were the lightness of the contribution, limited ablations/error analysis, uncertainty about generality/hyperparameters, figure clarity, and whether the scaffold hurts other benchmarks.

In rebuttal, the authors added systematic ablations on line count and thickness showing robustness, analyzed failure modes via line-object interference and spatial entropy, and reported broader results on MMBench/PHYBench and additional statistical tests. They also evaluated visual analogies and showed a small improvement on RAVEN by prompting models to use the existing grid structure, and noted minor compute overhead versus CoT. These additions materially strengthen the paper and address most factual/methodological concerns.

After discussion, two reviewers increased to borderline-accept and one remained positive; the remaining reservations concern conceptual simplicity and presentation polish rather than correctness. There is no fundamental flaw: the effect is real (replicated across models/tasks), limitations are now documented (e.g., interference/high clustering), and the scope is appropriately framed as a practical input-design lever rather than a new learning algorithm.

Given all this, the AC recommends the paper be accepted, with a request to incorporate the new analyses, clarify baselines/naming, tighten figures/tables, and make the limitations/failure modes prominent in the camera-ready.